



# The stratospheric Brewer–Dobson circulation inferred from age of air in the ERA5 reanalysis

Felix Ploeger[1,2], Mohamadou Diallo[1], Edward Charlesworth[1], Paul Konopka[1], Bernard Legras[3], Johannes C. Laube[1], Jens-Uwe Grooß[1], Gebhard Günther[1], Andreas Engel[4], and Martin Riese[1]

[1]Institute for Energy and Climate Research: Stratosphere (IEK–7), Forschungszentrum Jülich, Jülich, Germany.
[2]Institute for Atmospheric and Environmental Research, University of Wuppertal, Wuppertal, Germany.
[3]Laboratoire de Météorologie Dynamique, UMR8539, IPSL, UPMC/ENS/CNRS/Ecole Polytechnique, Paris, France.
[4]Institute for Atmospheric and Environmental Sciences, Goethe-University Frankfurt, Frankfurt, Germany.

**Correspondence:** Felix Ploeger (f.ploeger@fz-juelich.de)

**Abstract.** This paper investigates the global stratospheric Brewer-Dobson circulation (BDC) in the ERA5 meteorological reanalysis from the European Centre for Medium-Range Weather Forecasts (ECMWF). The analysis is based on simulations of stratospheric mean age of air, including the full age spectrum, with the Lagrangian transport model CLaMS, driven by winds and total diabatic heating rates from the reanalysis. ERA5-based results are compared to those of the preceding ERA–Interim reanalysis. Our results show a significantly slower BDC for ERA5 than for ERA–Interim, manifesting in weaker diabatic heating rates and larger age of air. In the tropical lower stratosphere, heating rates are 30–40% weaker in ERA5, likely correcting a known bias in ERA–Interim. Above, ERA5 age of air appears slightly high-biased and the BDC slightly slow compared to tracer observations. The age trend in ERA5 over 1989–2018 is negative throughout the stratosphere, as climate models predict in response to global warming. However, the age decrease is not linear over the period but exhibits steplike changes which could be caused by muti-annual variability or changes in the assimilation system. Over the 2002–2012 period, ERA5 age shows a similar hemispheric dipole trend pattern as ERA–Interim, with age increasing in the NH and decreasing in the SH. Shifts in the age spectrum peak and residual circulation transit times indicate that reanalysis differences in age are likely caused by differences in the residual circulation. In particular, the shallow BDC branch accelerates similarly in both reanalyses while the deep branch accelerates in ERA5 and decelerates in ERA–Interim.

## 1 Introduction

The Brewer-Dobson circulation (BDC) is the global transport circulation in the stratosphere, controlling the transport of chemical species and aerosol (e.g., Holton et al., 1995; Butchart, 2014). Changes in the BDC induce changes in radiatively active trace gas species and hence may cause radiative effects on climate. Therefore, BDC changes need to be reliably represented in atmospheric models.

The BDC is characterized by upwelling motion in the tropics, poleward transport in the stratosphere and downwelling above middle and high latitudes. As pointed out by Haynes et al. (1991), this circulation is mechanically driven by atmospheric waves propagating upwards from the troposphere and breaking at upper levels in the stratosphere where they deposit their momentum



and cause the driving force. From a conceptual point of view, the BDC can be divided into a residual mean mass circulation and additional two-way eddy mixing (e.g., Neu and Plumb, 1999; Garny et al., 2014), which are both related to the breaking

of atmospheric waves. The residual circulation and eddy mixing both affect trace gas distributions in a complex manner (e.g., Minganti et al., 2020). The climatological structure of the BDC shows two main circulation branches: a shallow branch at lower levels (below about 20 km) which causes rapid transport to high latitudes (transport time scales of months), and a deep branch above with much longer transport time scales of a few years (e.g., Plumb, 2002; Birner and Bönisch, 2011; Lin and Fu, 2013).

Diagnosing the BDC and estimating its strength is a challenging task due to the fact that the BDC is a zonal mean circulation and that the mean vertical velocities are very slow (less than 1 mm/s). In models, these slow velocities can be computed (e.g., within the Transformed Eulerian Mean (TEM) framework, Andrews et al., 1987), but are likely affected by model numerics. For observations, the circulation strength needs to be deduced from trace gas measurements. A common diagnostic for the speed of the BDC is the stratospheric age of air (e.g., Waugh and Hall, 2002). Due to atmospheric mixing processes, a given

air parcel in the stratosphere is characterized by a multitude of different transit times through the stratosphere, defining the age spectrum (Hall and Plumb, 1994). The first moment of the age spectrum defines the mean age of air. Due to its definition of being an average transit time, mean age may give ambiguous results, while the age spectrum is able to resolve the information of different processes (e.g., Waugh and Hall, 2002).

Despite its crucial effects on atmospheric composition and climate no common understanding of long-term changes of the

BDC with increasing greenhouse gas levels has been reached yet. On the one hand, climate models show a robust strengthening and acceleration of the BDC with climate change (e.g., Butchart et al., 2010), manifesting in an increase in tropical upwelling and a decrease in global mean age of air. On the other hand, atmospheric trace gas measurements from balloon soundings at NH mid latitudes show a non-significant long-term BDC trend over the last decades (Engel et al., 2009, 2017; Fritsch et al., 2020). Also satellite measurement from the Michelson Interferometer for Passive Atmospheric Sounding (MIPAS) show no

globally homogeneous mean age change over the 2002–2012 period but a more detailed pattern with increasing age in the Northern hemisphere (NH) and decreasing age in the Southern hemisphere (SH) (Stiller et al., 2012; Haenel et al., 2015).

Coming along with improvements in model physics and data assimilation systems, meteorological reanalyses have been used more intensively for trend investigations during recent years. With transport driven by ECMWF ERA–Interim reanalysis modelling studies have shown a weak increase of mean age in the NH middle stratosphere, qualitatively consistent with balloon

observations (e.g., Diallo et al., 2012; Monge-Sanz et al., 2012). However, more recent efforts combining different newest generation reanalysis data sets have shown a large dependency of BDC trend estimates on the reanalysis used, for both residual circulation (Abalos et al., 2015; Miyazaki et al., 2016) and age of air diagnostics (Chabrillat et al., 2018; Ploeger et al., 2019). Hence, while inter-annual BDC variability seems to be well represented in reanalyses, regarding decadal changes and trends no consensus has been reached amongst reanalyses, including ERA–Interim reanalysis from the European Centre for Medium-

Range Weather Forecasts (Dee et al., 2011), JRA–55 from the Japanese Meteorological Agency (Kobayashi et al., 2015), and MERRA–2 from the National Aeronautics and Space Administration (Gelaro et al., 2017).





Very recently, the ECMWF has released its newest generation reanalysis product ERA5 (Hersbach et al., 2020). Compared to its predecessor ERA–Interim, ERA5 is based on an improved forecast model version and improved data assimilation system (see Sect. 2.2 for further details). Case studies on stratospheric and tropospheric transport indicate improvements in the rep-
resentation of physical processes, like convection, in ERA5 (e.g., Li et al., 2020). Recently, Diallo et al. (2020) have analyzed the ERA5 residual mean mass circulation of the BDC, its variability and trend, based on Transformed Eulerian mean (TEM) calculations. The present paper can be viewed complementary as it investigates the representation of the BDC in ERA5 in terms of stratospheric age of air from transport model simulations, and compares results to the previous ERA–Interim reanalysis. For that reason, we carry out simulations of stratospheric age of air with the Lagrangian CLaMS model (Chemical Lagrangian
Model of the Stratosphere) over the period 1979–2018 driven with either ERA5 or ERA–Interim reanalysis meteorology. Simulation of the stratospheric age spectrum with CLaMS allows pinpointing differences between the reanalyses to processes. The main research questions are: (i) How strong and fast is BDC transport in ERA5 compared to ERA–Interim? (ii) How has the BDC (and age of air) been changing over recent decades? (iii) How good is the agreement with age of air derived from trace gas observations? The analysis presented here can be regarded as a follow-up of the analyses presented by Chabrillat et al. (2018)
and Ploeger et al. (2019), where the BDC in ERA–Interim, JRA–55 and MERRA–2 reanalysis was compared in terms of age of air, and we discuss the ERA5 results within the context of the other reanalyses. For better comparability, we also present ERA–Interim results from Ploeger et al. (2019) here, although for an extended period, and juxtapose them with the new ERA5 results.

In a first step in Sect. 2 the CLaMS model and age spectrum calculation is described, as well as ERA5 reanalysis data. A
particular focus is laid on the ERA5 diabatic heating rate which is used to drive CLaMS model transport. Thereafter, Sect. 3 presents the results related to the climatological structure of the BDC, showing that ERA5 has a substantially slower BDC than ERA–Interim. Section 4 presents BDC and age of air trends, showing a globally negative long-term trend for ERA5 and a stronger variability compared to ERA–Interim. A comparison to age of air observations is presented in Sect. 5. The results are placed into context of previous studies in the discussion in Sect. 6, and final conclusions are summarized in Sect. 7.

## 2  Data and method

In this study, the BDC is investigated based on age of air calculated with the CLaMS model driven with ERA5 and ERA–Interim reanalysis data. In the following, Sect. 2.1 describes the CLaMS model and age of air calculation while Sect. 2.2 briefly describes the ERA5 reanalysis with focus on the used variables.

### 2.1  Age of air simulations with the Chemical Lagrangian Model of the Stratosphere (CLaMS)

The CLaMS model is a Lagrangian chemistry transport model, with the transport scheme based on the calculation of three-dimensional air parcel trajectories (which represent the model grid points) and a parameterization of small-scale atmospheric mixing (McKenna et al., 2002; Konopka et al., 2004). This mixing parameterization is controlled by the shear in the large-scale flow (via a critical Lyapunov exponent), such that in regions of large flow deformations strong mixing occurs. The forward





trajectory calculation is driven with offline meteorological data. In this paper, we will use ERA5 and ERA–Interim reanalysis
data which are further described in Sect. 2.2.

In the vertical direction CLaMS uses an isentropic vertical coordinate which makes the model particularly well-suited for the
stratosphere, where diabatic transport is generally weaker than adiabatic transport. Strictly speaking, the vertical coordinate in
CLaMS is a hybrid potential temperature which is orography-following at the surface and transforming smoothly into potential
temperature above such that it equals $\theta$ in the stratosphere (e.g., Mahowald et al., 2002; Pommrich et al., 2014). The cross-
isentropic vertical velocity is calculated from the reanalysis total diabatic heating rate (see Sect. 2.2 for further details).

A calculation of the fully time-dependent stratospheric age of air spectra has been implemented in CLaMS, based on multiple
tracer pulses, as described by Ploeger et al. (2019), and references therein. The age spectrum $G(r,t,\tau)$ at a point $r$ in the
stratosphere and time $t$ is the distribution of transit times $\tau$ from the surface (or from the tropopause in some studies) to $r$. For
the age spectrum calculation in CLaMS, chemically inert model tracers are pulsed in the orography-following lowest model
layer (approximately the boundary layer) in the tropics (30°S–30°N) with unit mixing ratio, a pulse duration of 1 month and a
pulse frequency of 2 months. The value of the age spectrum at transit time $\tau_i$ is then related to the mixing ratio $\chi_i$ of the tracer
pulsed at $t - \tau_i$ and sampled at $r$ (e.g., Li et al., 2012)

$$G(r,t,\tau_i) = \chi_i \; . \tag{1}$$

Therefore, the use of 60 pulse tracers in the model with pulse frequency 2 months allows calculation of the age spectrum over
10 years along transit time (e.g., Ploeger et al., 2019). Mean age $\Gamma$ is the average stratospheric transit time and is defined as the
first moment of the age spectrum

$$\Gamma(r,t) = \int\limits_0^\infty \mathrm{d}\tau \, \tau \, G(r,t,\tau) \; . \tag{2}$$

The two model simulations driven with either ERA5 or ERA–Interim cover the period 1979–2018. Preceding 10 year spin-
up simulations have been carried out by repeating the meteorology of 1979. To eliminate influence of this spin-up on the age
spectra we analyze the period 1989–2018, in the following, when all memory of the spin-up in the 10 year long age spectra
is lost. As the model age spectrum is truncated at 10 years the respective mean age will be low biased if no correction for
the spectrum tail is taken into account. Therefore, throughout this paper we consider mean age calculated from an additional
"clock-tracer" in CLaMS, a chemically inert tracer with linearly increasing mixing ratios at the surface (e.g., Hall and Plumb,
1994). This clock-tracer mean age had experienced a longer spin-up (minimum 20 years, at the beginning of the considered
period in 1989), and is therefore larger compared to the spectrum-based mean age.

In addition to age of air, we also consider CFC–11 from CLaMS simulations in Sect. 5. The simulations include a simplified
chemistry scheme (Pommrich et al., 2014) such that CFC–11 is affected only by photolytical loss. Both mean age and CFC–11
simulated with CLaMS agree well with stratospheric observations (e.g., Ploeger et al., 2019; Laube et al., 2020). Note that
compared to Ploeger et al. (2019) small quantitative differences can occur due to the use of different mean age calculation
methods (clock-tracer versus spectrum-based), and also due to different periods considered. Another difference to the simula-
tions by Ploeger et al. (2019) concerns the cross-isentropic vertical velocities. Ploeger et al. (2019) added a constant correction





term to the vertical velocity to correct for missing balance in the annual mean cross-isentropic mass flux on a given $\theta$-level, as suggested by Rosenlof (1995) and implemented in CLaMS by Konopka et al. (2010). Here, we do not include this mass correction and use the "raw" reanalysis heating rate to simplify interpretation and reproducibility with other studies. For most

of the results including this annual mean mass balance causes no significant change. Only for mean age trends after about 2002 it causes a clearer age decrease in the SH (see Sect. 4).

## 2.2 ERA5 reanalysis

The newest generation ERA5 reanalysis from the European Centre for Medium-Range Weather Forecasts (ECMWF) is the successor of the previous ERA–Interim reanalysis (Dee et al., 2011). ERA5 is now available from 1979–2020 with production

lagging real time by about 2 months. For the production of ERA5, 4D–Var data assimilation of the ECMWF Integrated Forecast System (in cycle CY41R2) was used. The horizontal resolution is about 30 km (T639). In the vertical, the pressure range from the surface to 0.01 hPa is covered with 137 hybrid levels. The output frequency for ERA5 data is hourly. Due to a cold bias in the lower stratosphere, the reanalysis data has been replaced for the period 2000–2006 with the updated data named ERA5.1. For the present paper, we carried out CLaMS simulations driven with both the previous (termed ERA5.0 in the following)

and the corrected ERA5.1 data (for simplicity, termed ERA5 in this paper), and discuss effects of the bias correction on the stratospheric BDC (Sect. 6). Further details on ERA5 can be found in Hersbach et al. (2020).

For better comparability between the ERA5 and ERA–Interim driven model simulations as well as for practicability reasons (storage and memory space) we use 6-hourly (0, 6, 12, 18 UTC) ERA5 data with truncated $1 \times 1$ degree horizontal resolution, as provided by the ECMWF. However, we maintain the full vertical resolution. Hence, the ERA5 data to drive the CLaMS

model in this study has 137 hybrid ECMWF model levels, for ERA–Interim 60 levels.

Of particular importance for the CLaMS model calculations is the reanalysis temperature tendency variable which is used for deducing diabatic vertical velocity (diabatic heating rates). Both ERA5 and ERA–Interim provide 5 temperature tendencies: The mean temperature tendencies due to short-wave and long-wave radiation for both clear-sky and all-sky conditions, as well as the mean temperature tendency due to parametrisations. From the temperature tendency due to parametrisations (the total

diabatic heating rate $Q_\mathrm{tot}$) the cross-isentropic vertical velocity $\mathrm{d}\theta/\mathrm{d}t$ for driving CLaMS transport is calculated as

$$\frac{\mathrm{d}\theta}{\mathrm{d}t} = \frac{\theta}{c_p T} Q_\mathrm{tot} , \tag{3}$$

with $T$ temperature, $\theta$ potential temperature, and $c_p$ the specific heat capacity at constant pressure. The calculation of cross-isentropic vertical velocity for ERA–Interim was described by Ploeger et al. (2010). In the following, we illustrate the similar (but not identical) procedure for ERA5.

Temperature tendencies are only available from the ERA5 forecast data, which is stored twice per day (6, 18 UTC) with forecast steps ranging between 1 and 18 hours (1 hour increments). These temperature tendencies have to be interpreted differently compared to ERA–Interim, as they are "mean rates", representative for the interval between the actual time and the previous post processing time. For instance, the forecast data at 6 UTC and 5 hour forecast step is the mean tendency between 10 and 11 UTC. Temperature tendencies are provided in K/s (Kelvin per second). The CLaMS pre-processor assigns





the forecast variables to the reanalysis data set at the later time (here 11 UTC). This induces a time uncertainty of 0.5 hours, which is negligible for the purpose of this paper.

In cases where not the hourly ERA5 data is used to drive model transport but data sub-sampled in time, the temperature tendencies have to be averaged appropriately, as they represent accumulations since the previous post processing time and not since the forecast date. Otherwise, sampling errors will occur. Hence, for the present case of using only 0, 6, 12, 18 UTC data

we average all tendency data sets within 6 hour windows around each date. For instance, the mean tendency averaged over 10-15 UTC (from forecast data at 6 UTC with 4-9 hour steps, see above) is assigned to the reanalysis data at 12 UTC.

## 3    Climatological view on the stratospheric circulation

As described in Sect. 2, CLaMS uses diabatic heating rates for driving vertical transport in the model. The climatological cross-isentropic velocity $\mathrm{d}\theta/\mathrm{d}t$ calculated from the diabatic heating rate for winter and summer seasons from ERA5 and

ERA–Interim is shown in Fig. 1 a–d. Overall, both reanalyses show very similar distributions and seasonality for $\mathrm{d}\theta/\mathrm{d}t$. Cross-isentropic tropical upwelling maximizes in boreal winter, somewhat shifted into the summer subtropics, and the extratropical downwelling maximizes in the respective winter hemisphere. Further, heating rates show a double-peak structure above the tropical tropopause with a minimum above the equator. However, clear quantitative differences occur between the two reanalyses. The zoomed view in Fig. 1 e–g reveals that in the layer 400-430 K (30°N/S) ERA–Interim shows about 40% larger total

diabatic heating rates than ERA5. Furthermore, at lowest TTL levels around the level of zero radiative heating (about 350 K) total diabatic heating rates are much weaker in ERA5 than ERA–Interim, with even no continuous upwelling in the annual mean in a shallow layer around 350 K (Fig. 1g). It should however be noted, that seasonal ERA5 heating rate averages show very weak zonal mean upwelling from the troposphere into the stratosphere in small latitude bands, which vanishes in the annual mean due to seasonal shifts. This much weaker upwelling in the TTL and tropical lower stratosphere causes stronger

restrictions on large-scale advective upward transport in ERA5. The heating rate difference below the tropopause will also affect the simulations of mean age, which is defined with respect to the surface in this study. Another difference between the two reanalyses concerns heating rates in the summertime stratosphere above about 800 K (about 20 hPa), where ERA5 shows upward whereas ERA–Interim shows downward velocities. The upward velocities in ERA5 in that region are more consistent with the residual circulation vertical velocity (see next paragraph and Fig. 2).

Diallo et al. (2020) analyzed the ERA5 BDC based on TEM residual circulation vertical velocity, a common measure for the strength of the BDC (e.g., Andrews et al., 1987, Eq. 3.5.1b),

$$\overline{w}^* = \overline{w} + \frac{1}{a\cos\phi}\,\partial_\phi\left(\cos\phi\,\frac{\overline{v'\theta'}}{\partial_z\overline{\theta}}\right)\,,\tag{4}$$

where the overline denotes the zonal mean and primes fluctuations therefrom (due to eddies), $w$ is the vertical velocity in log-pressure $z$ coordinates, $a$ is the Earth's radius, $\phi$ latitude, and $v$ is the latitudinal wind component. Here, we briefly recapitulate

a few findings from that paper for ease of comparison to the heating rate based results.







**Figure 1.** Total diabatic heating rate $d\theta/dt$ climatology (1979–2018) from (a) ERA5 during boreal winter (December–February, DJF) and (b) during summer (June–August, JJA). (c, d) Same but from ERA–Interim. (e, f) Zoomed view into the tropical lower stratosphere on annual mean heating rates, and (g) the respective difference ERA–Interim minus ERA5 (thin black countour in the difference plot shows the zero contour from ERA–Interim). Heating rate contours are highlighted in black (dashed for negative values). White contours show zonal wind (in ±10 m/s steps; easterly wind dashed), thin dashed black lines pressure levels, the thick black line the (WMO lapse rate) tropopause.

Figure 2 shows $\overline{w}^*$ calculated from ERA5 and ERA–Interim during boreal winter (December–February, DJF) and summer (June–August, JJA), for comparison with the diabatic heating rates. Overall, the $\overline{w}^*$ distributions and seasonal differences for both reanalyses are very similar. Also for $\overline{w}^*$, both reanalyses show strongest tropical upwelling in boreal winter, somewhat shifted into the summer subtropics, and strongest downwelling in the respective winter hemisphere. In particular in the winter





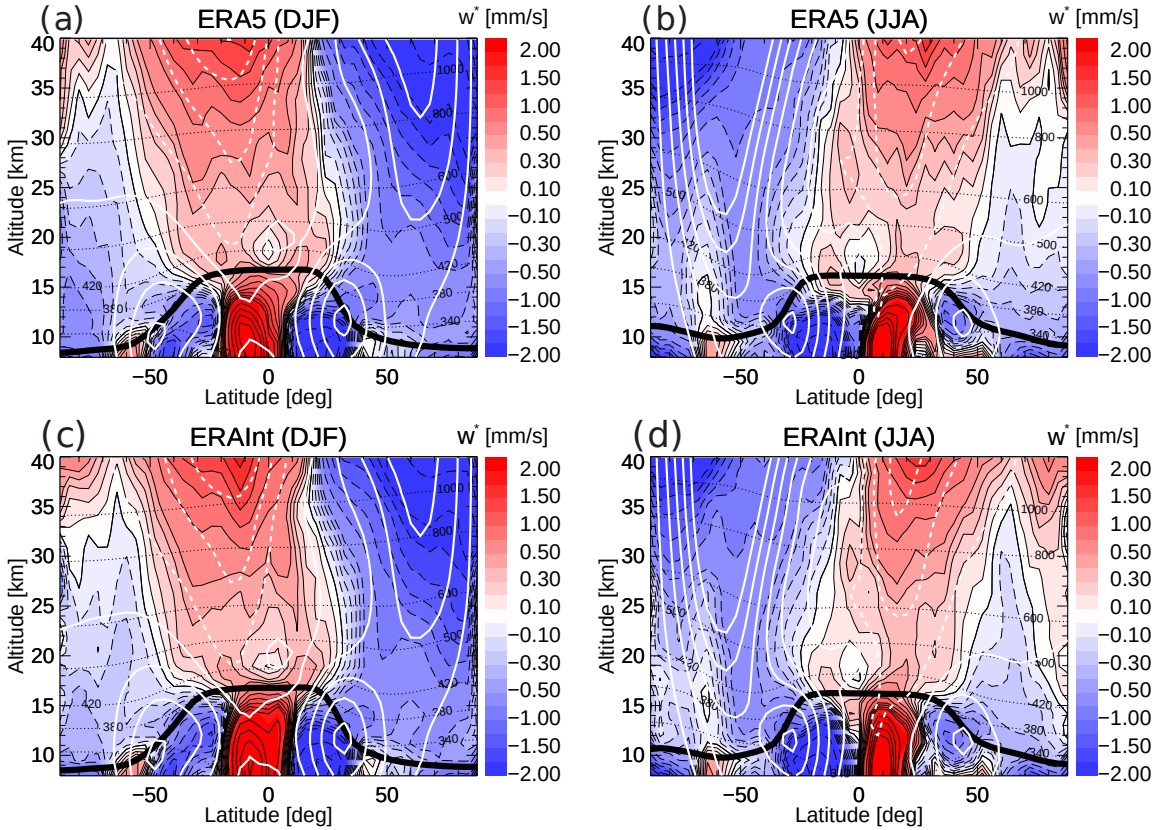

**Figure 2.** Residual circulation vertical velocity $\overline{w}^*$ climatology (1979–2018) from (a) ERA5 during boreal winter (December–February, DJF) and (b) during summer (June–August, JJA). (c, d) Same but from ERA–Interim. Circulation contours are highlighted in black (dashed for negative values). White contours show zonal wind (in $\pm 10$ m/s steps; easterly wind dashed), thin dashed black lines potential temperature levels, the thick black line the (WMO lapse rate) tropopause.

hemisphere, contours of negative $\overline{w}^*$ representing downwelling are very close for both reanalyses. Also in the deep tropics close to the equator around the 18 km level (about 420 K potential temperature, 70 hPa pressure), a minimum in upwelling (even downwelling in JJA) occurs similarly in both data sets. Minor differences between ERA5 and ERA–Interim concern downward velocities in the summer hemisphere and a stronger upwelling in the tropical lower stratosphere in ERA–Interim. Diallo et al. (2020) showed in their Fig. 2 that tropical upwelling differences in $\overline{w}^*$ in the tropical lower stratosphere amount

to about 40% at around 15 km and decrease above (zero difference at about 22 km). Hence, in the tropical lower stratosphere reanalysis differences in upwelling as diagnosed from $\overline{w}^*$ are similar to those diagnosed from heating rates (up to 30-40%, see Fig. 1). Above about 20 km, however, upwelling from heating rates shows larger differences than upwelling from $\overline{w}^*$. Downwelling differences are larger in the summer than winter hemisphere, similar to the case of $d\theta/dt$.





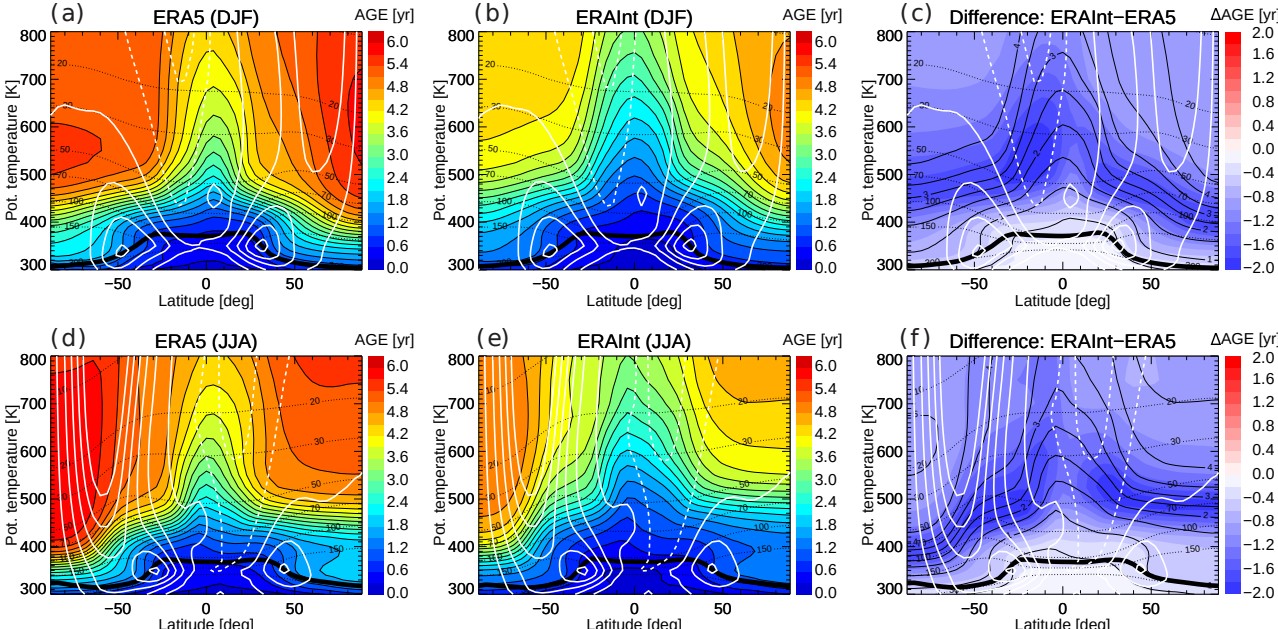

**Figure 3.** Mean age climatology (1989–2018) for boreal winter (December–February, DJF) from ERA5 (a), ERA–Interim (b), and the difference (c). (d, e, f) Same, but for boreal summer (June–August, JJA). White contours show zonal wind (in ±10 m/s steps; easterly wind dashed), thin dashed black lines pressure levels, the thick black line the (WMO lapse rate) tropopause.

The integrated effect of BDC transport in the two reanalyses is shown from climatological mean age for winter and summer
in Fig. 3. The general characteristics of the stratospheric mean age distribution are evident for both reanalyses, with age increasing with both altitude and latitude, low age in the tropical upwelling region and high age in extratropical downwelling regions. Oldest air is always found in the winter hemisphere, even older in the SH during JJA compared to NH during DJF. The summertime lowest stratosphere (below about 450 K) is characterized by a "flushing" with young air (e.g., Hegglin and Shepherd, 2007; Bönisch et al., 2009), strongest in the NH during JJA. A particular feature in this region is an age inversion in the summertime NH with younger air (around 400 K) located above older air (around 350 K), which is also apparent in both reanalysis. This "eave structure" (i.e., younger air above older air) in the summertime lower stratosphere age distribution has been recently discussed by Charlesworth et al. (2020) and has been shown to depend critially on the numerics of the model transport scheme.

Clear quantitative differences between ERA5 and ERA–Interim age are evident in Fig. 3c and f, with significantly older air for ERA5. Largest differences of up to about 2 years (equivalent to about 50–75%) occur in the lower stratospheric regions where the climatological age distribution (black contours) shows strongest gradients. These differences indicate a downward shift of the regions of strongest age gradients and stronger gradients in ERA5 compared to ERA–Interim.

To gain deeper insight into transport processes and their differences in the two reanalyses, we consider stratospheric age spectra at 400 K and 600 K potential temperature levels, representative for the shallow and the deep BDC branch, respec-





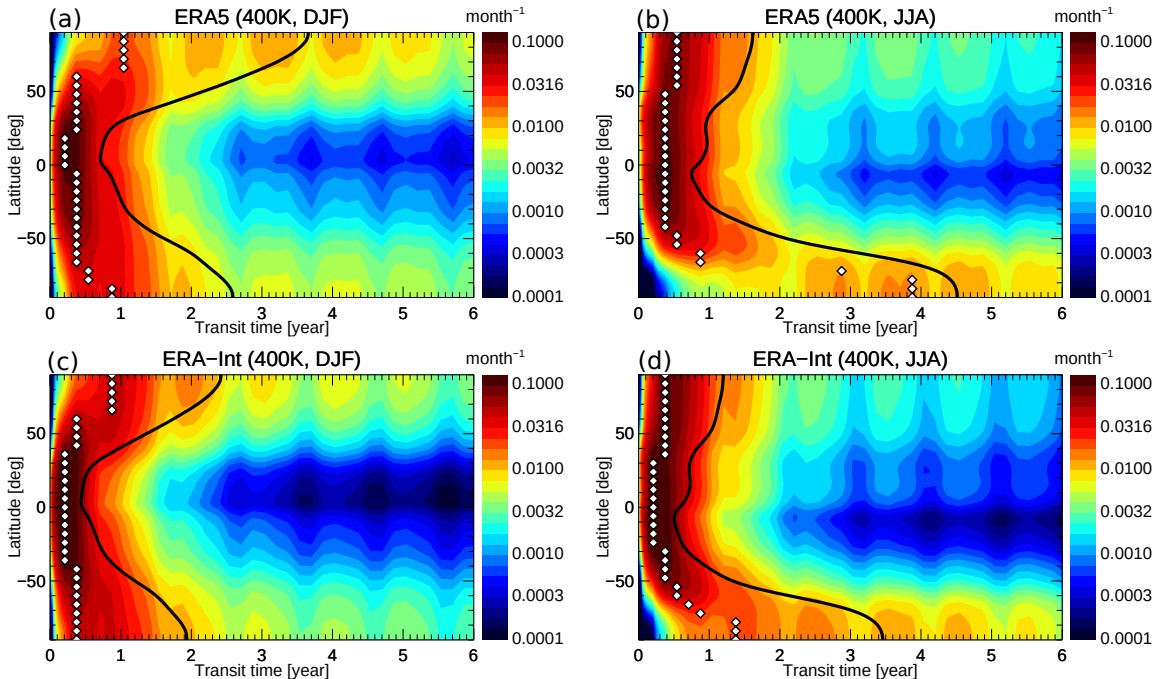

**Figure 4.** Age spectra at 400 K (1989–2018 climatology) from ERA5 for boreal winter (a) and summer (b). (c, d) Same, but for age spectra from ERA–Interim. Black contour shows mean age (calculated as first moment of spectra), white diamonds show modal age (peak of the spectrum).

tively (see Fig. 2 for a relation between potential temperature and altitude levels). Figure 4 shows the CLaMS model age

spectra at 400 K versus latitude for ERA5 and ERA–Interim for winter and summer. The spectra from both reanalyses show similar overall characteristics and seasonality, caused by known characteristics of BDC transport. The multi-modal spectrum shape, arising from the seasonality in upward transport into the stratosphere and in the strength of transport barriers, clearly emerges for both cases and appears strongest for middle latitude and high latitude spectra, consistent with previous studies (e.g., Reithmeier et al., 2007; Li et al., 2012). In the tropics, age spectra are narrower and with a younger peak in boreal winter,

indicating faster upwelling during that season. Weaker latitudinal gradients in the summer hemisphere, especially in the NH during JJA, are a sign of stronger mixing in summer. The flushing of the summertime lower stratosphere with young air can be seen from the extent of the young air peak to high latitudes in the summer hemisphere (e.g., Fig. 4b, d).

Closer inspection reveals also clear differences between ERA5 and ERA–Interim age spectra. In the tropics, the main spectrum peak is broader and shifted to larger transit times for ERA5, related to slower tropical upwelling compared to ERA–

Interim. At high latitudes this shift of ERA5 age spectra towards older transit times is even clearer, indicating slower transport along the BDC and a stronger confinement of polar regions against exchange with middle latitudes in ERA5. Clearest differences occur in the SH polar vortex during winter (JJA) where the modal age (transit time of the largest spectrum peak) for





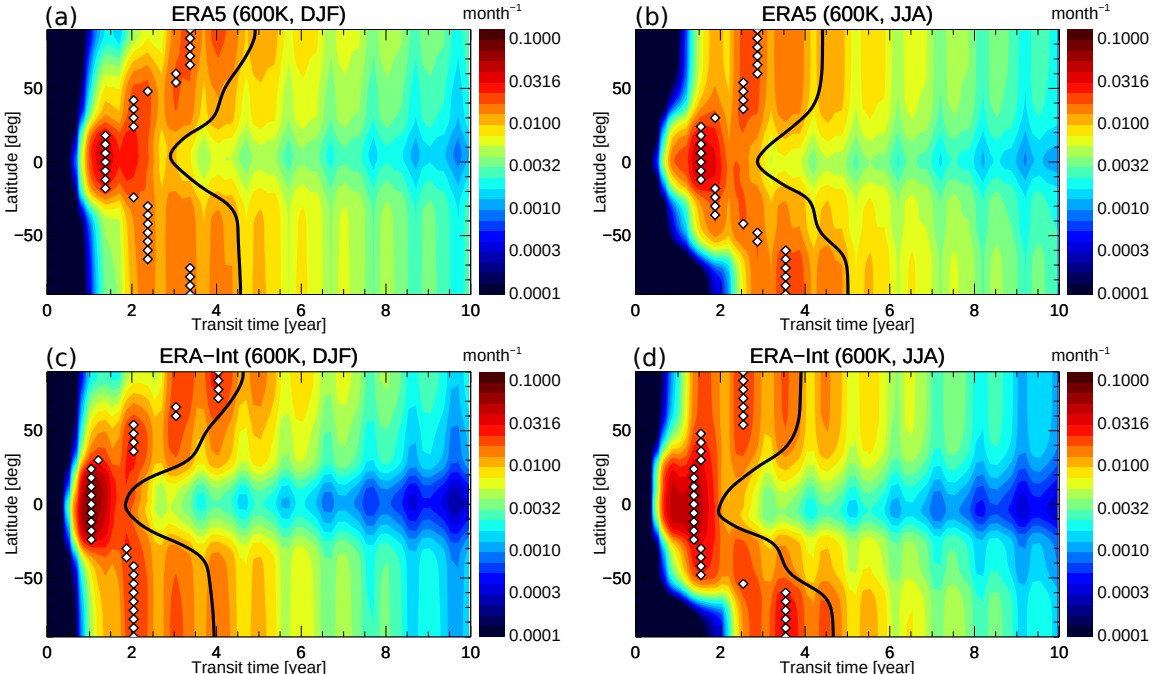

**Figure 5.** Same as Fig. 4 but at 600 K.

ERA5 is at about 4 years, while for ERA–Interim modal age occurs at 1.4 years. Hence, there is a significantly higher fraction of young air at high latitudes for ERA–Interim than for ERA5.

Similar conclusions hold for the age spectrum comparison at 600 K (Fig. 5). While the general spectrum characteristics are similar for ERA5 and ERA–Interim, clear detailed differences occur in the spectrum shape. ERA5 spectra are shifted towards older transit times and show an older peak compared to ERA–Interim, as found similarly at the lower 400 K level. In the tropics, these differences are particularly clear with the shift of the spectrum peak to older ages in ERA5 compared to ERA–Interim indicating slower residual circulation upwelling (e.g., Li et al., 2012; Ploeger et al., 2019). This slower residual circulation

upwelling is consistent with the weaker diabatic heating rates in the TTL and lower tropical stratosphere in Fig. 1. Moreover, the larger extent of the youngest spectrum peak towards high latitudes in the summer hemisphere (e.g., in the NH during JJA, Fig. 5b) shows faster transport of young air towards the pole in ERA–Interim than in ERA5.

## 4    Circulation and age changes over (multi-)decadal time scales

Trends in mean age over the entire 30 year period 1989–2018 and over 2002–2012 are shown in Fig. 6. The trends are calculated

from linear regression of the deseasonalized time series (after subtracting the mean annual cycle at each grid point). Even over 30 years trends values are still affected by decadal variability, as will be discussed at the end of this section. Nevertheless, for





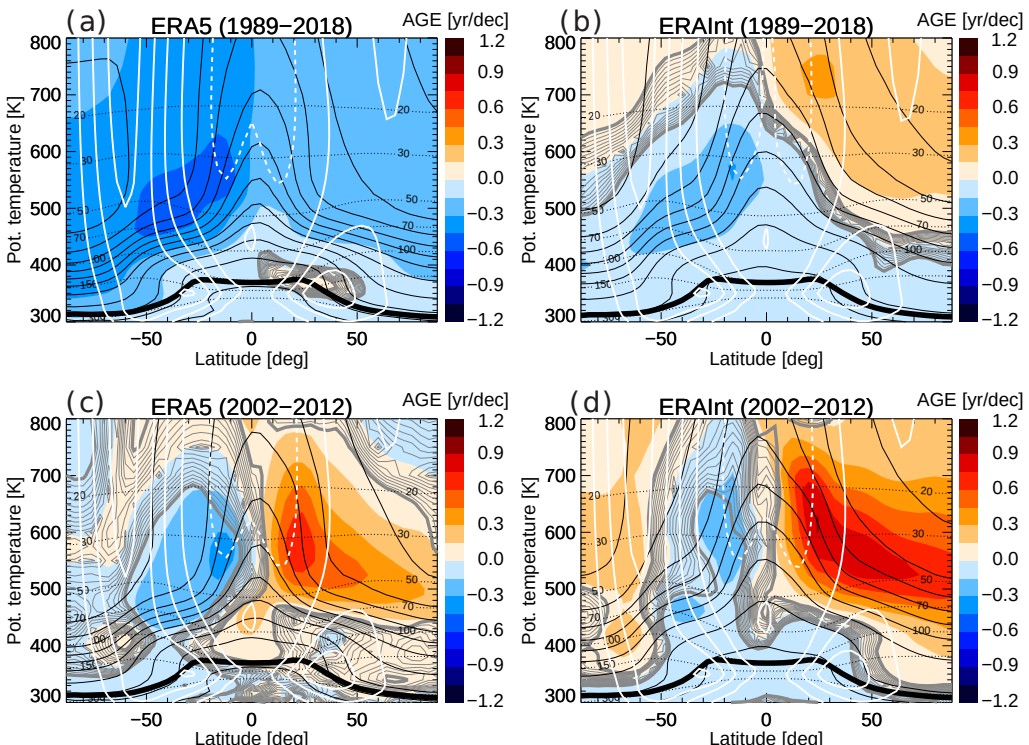

**Figure 6.** Mean age trends for the period 1989–2018 from ERA5 (a) and ERA–Interim (b). (c–d) Same, but for the 2002–2012 period. Black contours (solid) show climatological mean age, thin dashed black lines pressure levels, white contours zonal wind (in $\pm 10$ m/s steps; easterly wind dashed), and the thick black line the tropopause. The significance of the linear trend, measured in multiples of the standard deviation $\sigma$, is shown as gray contours ($2\sigma$ contour as thick, then decreasing with 0.2 step as thin lines).

simplicity we use the term "trend" in this paper but note that our results concern changes over 30 years and not over centennial time scales, as is often the case for climate model experiments.

Evidently, ERA5 shows a negative age trend throughout the stratosphere over the longer period, with strongest decreases of
up to $-0.5$ years per decade (yr/dec) in the SH sub-tropics and mid latitudes. In the SH and in the lowest stratosphere (below about 450 K) this negative age trend qualitatively agrees with the trend from ERA–Interim. In particular in the NH above about 500 K, however, the signs of the trends in the two reanalyses are opposite, with ERA–Interim showing increasing age. These differences will be further discussed in Sect. 6.

For the shorter period 2002–2012 both reanalyses show qualitatively similar mean age changes, with increasing age in the
NH and decreasing age in the SH (Fig. 6c–d). Detailed differences concern a weaker NH age increase and a stronger SH age decrease in ERA5 compared to ERA–Interim. In the lowest tropical and sub-tropical stratosphere ERA5 shows increasing age, although non-significant changes in some regions, compared to decreasing age in ERA–Interim in this region. Hence, changes in the shallow BDC branch over this short period are not consistent between the two reanalyses. The small quantitative





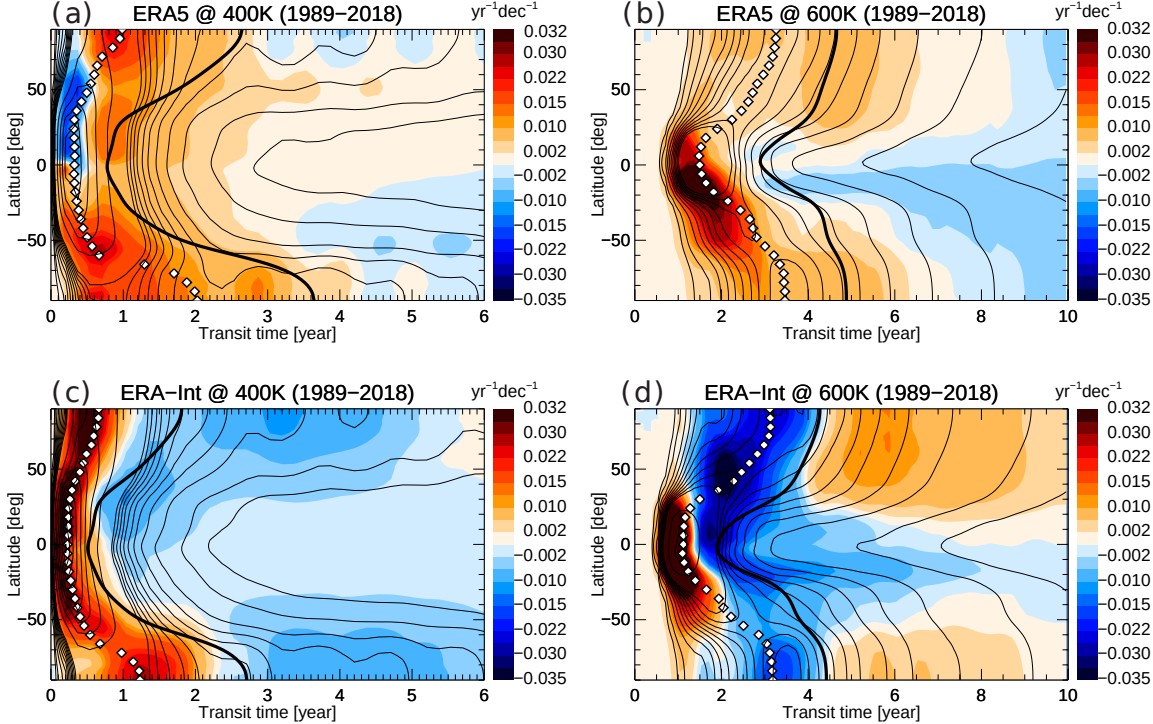

**Figure 7.** Age spectrum trend for the period 1989–2018 from ERA5 at 400 K (a), and 600 K (c). (c, d) Same, but for ERA–Interim. Thin black contours show annual mean age spectra, thick black contour shows mean age (calculated as first moment of spectra), white diamonds show modal age (peak of the spectrum). Note the different x-axis scales for the two levels.

differences for the 2002–2012 age trends compared to recent publications of CLaMS simulated mean age (e.g., Stiller et al., 2017) are related to the updates in the model configuration (e.g., exclusion of annual mean cross-isentropic mass balance here) and the use of clock-tracer versus age spectrum based mean age, as explained in Sect. 2.1.

Trends in the age spectrum provide more detailed information about changes in transport processes and are presented in Fig. 7 for the period 1989–2018 (for the two levels 400 and 600 K). ERA5 age spectra show a shift of the spectrum peak towards younger age for most regions, as indicated by relatively stronger positive spectrum trends at transit times shorter than 260 the modal age (indicating an increase in the mass fraction of young air). Such a decrease of modal age can be interpreted as an acceleration of the residual circulation, at least in the tropics and in the winter hemisphere extratropics (e.g., Li et al., 2012; Ploeger et al., 2019). In the tropics and SH this modal age shift and residual circulation acceleration is clearest. In the NH in a shallow layer around the 400 K level between 0–50°N the age changes are different, with a decrease of the young air fraction (transit times shorter than modal age) causing weak positive mean age changes (compare Fig. 7a and Fig. 6a).

The clearest difference to ERA–Interim regarding structural age spectrum changes emerges at middle and high latitudes at upper levels (here 600 K, Fig. 7b and d). ERA5, on the one hand, shows a shift of the spectrum to younger ages, although not


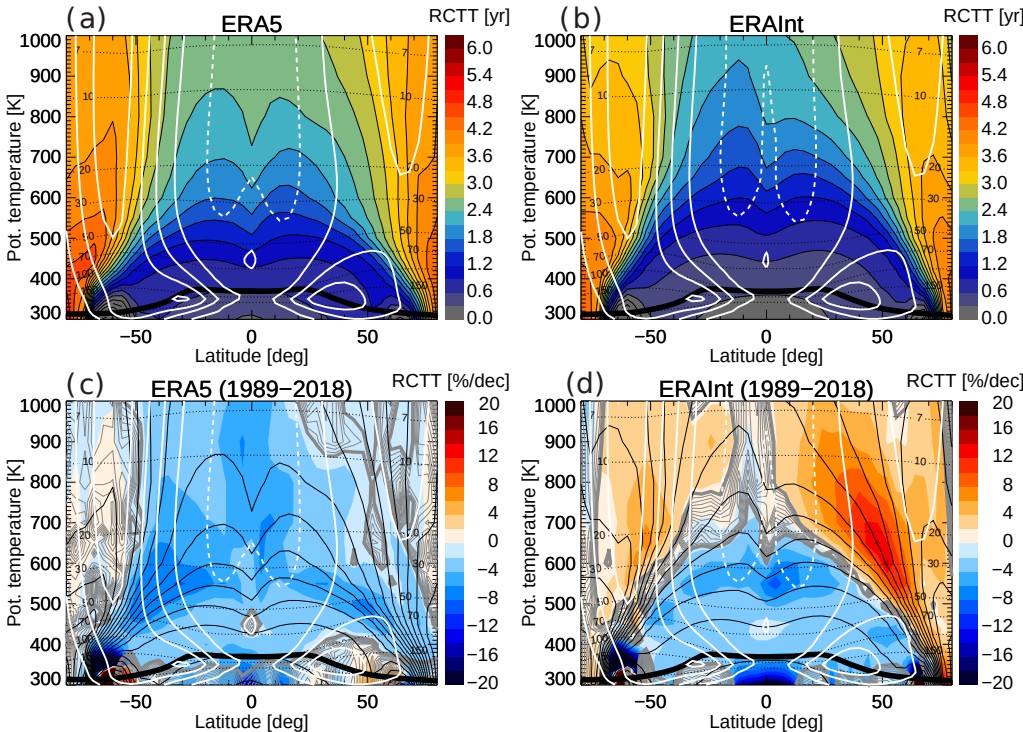

**Figure 8.** Residual circulation transit times (RCTT) from ERA5 (a) and ERA–Interim (b), and their trends over 1989–2018 (c, d). Thin black lines in c and d show climatological RCTT contours (in 0.3 yr steps). White contours show zonal wind (in ±10 m/s steps; easterly wind dashed), thin dashed black lines potential temperature levels, the thick black line the (WMO lapse rate) tropopause. The significance of the linear trend, measured in multiples of the standard deviation $\sigma$, is shown as gray contours (2 $\sigma$ contour as thick, then decreasing with 0.2 step as thin lines).

as clear as in the tropics and in the SH. ERA–Interim, on the other hand, shows a decrease in the fraction of air younger than about 4 years and an increase of the fraction of older air. This increased fraction of air older than about 4 years in ERA–Interim indicates a strengthening in the deep branch of the residual circulation. The different spectrum changes in ERA5 and ERA–

Interim cause the opposite mean age changes in the two reanalyses (Fig. 6), and are related to different trends in the deep BDC branch (see next paragraph).

    Changes in the structure of the residual circulation are further investigated by using residual circulation transit times (RCTTs), the pure transit time for (hypothetical) air parcels in the 2D residual circulation flow (e.g., Birner and Bönisch, 2011). Here, the RCTTs have been calculated in isentropic coordinates using the same reanalysis diabatic heating rates as in

the full CLaMS simulation for calculating vertical motion (e.g., Ploeger et al., 2019). Differences between mean age and RCTT are related to mixing effects (e.g., Garny et al., 2014; Dietmüller et al., 2017). Figure 8 shows the climatology and percentage trend in RCTT for the period 1989–2018 for ERA5 and ERA–Interim. Comparison of climatological RCTTs shows substan-





tially longer transit times for ERA5 than ERA–Interim (up to 40% longer in the lower stratosphere), consistent with the slower circulation in ERA5 as already diagnosed from heating rates, $\overline{w}^*$ and age of air. The trends in RCTTs in Fig. 8c–d indicate
differences between the reanalyses regarding changes in the structure of the BDC. In the lower stratosphere below about 600 K both reanalyses show consistent changes, with decreasing RCTT indicating a strengthening of the shallow residual circulation branch. In the tropical lower stratosphere the residual circulation accelerates by about 2.4 %/decade in ERA5 and 2.2 %/decade in ERA–Interim, as diagnosed from the RCTT trend (20°N/S, 450 K average, approximately 70 hPa). Clear differences emerge at higher levels and also at higher latitudes, hence in atmospheric regions of the deep circulation branch. In these regions, ERA5
still shows decreasing RCTTs, turning into insignificant changes at high latitudes. ERA–Interim, on the other hand, shows increasing RCTTs, clearest in the NH. Hence, changes in the deep branch of the residual circulation clearly differ between the reanalyses, with a weakly strengthening deep branch in ERA5 and a weakening deep branch in ERA–Interim.

Figure 9 provides further insights into the temporal evolution of mean age at three different locations in the stratosphere. The three locations have been chosen to be representative for the tropical lower stratosphere, and the NH and SH subtropical
stratosphere regions of strong mean age trends (compare Fig. 6). In the tropical lower stratosphere (450 K and 10°S/N; Fig. 9a), the relative difference between ERA5 and ERA–Interim is large and the variability in ERA5 mean age is significantly stronger than in ERA–Interim. At the higher levels, the variability in mean age in the two reanalyses is more comparable in magnitude. From a qualitative point of view, the variability is similar at all locations, with coinciding ups and downs in the age time series, e.g. caused by modulations in the BDC related to the Quasi-Biennial Oscillation (QBO) or El Niño Southern
Oscillation ENSO (e.g., Calvo et al., 2010; Konopka et al., 2016). A particularly striking feature is the anomalously high mean age following the year 1991, even higher for ERA5 than ERA–Interim. This significant increase in stratospheric mean age in reanalyses has been related to the Mt. Pinatubo eruption in June 1991 (Diallo et al., 2017). A more detailed analysis of mean age variability is left for future studies.

While Fig. 6 suggests a clear decrease in mean age throughout the global stratosphere, the time series in Fig. 9 show that
this decrease is not a simple linear trend over the 30 years considered. In fact, outside the tropics mean age appears to increase before about 1991 and after about the year 2000 (except in the SH), and decreases in between. These steplike changes are evident in both reanalyses and could be related to true atmospheric variability or to changes in the reanalysis assimilation system. In particular in the NH (Fig. 9c), the negative age trend in ERA5 during 1989–2018 appears to be related to the strongly increased age values at the beginning of the period.

## 5   Comparison to trace gas observations

Climatological reanalysis mean age at 20 km altitude is compared to mean age from $CO_2$ and $SF_6$ trace gas measurements in Fig. 10. Comparison is made to data from in-situ observations (as compiled by Waugh and Hall, 2002, for a detailed data description see the references therein). Mean age deduced from observed $SF_6$ is higher than mean age deduced from $CO_2$, consistent with the existence of a significant $SF_6$ chemical sink in the mesosphere (Ray et al., 2017). ERA–Interim agrees well
with the in-situ data (best for $CO_2$-based mean age) while the higher ERA5 mean age values are just at the upper margin





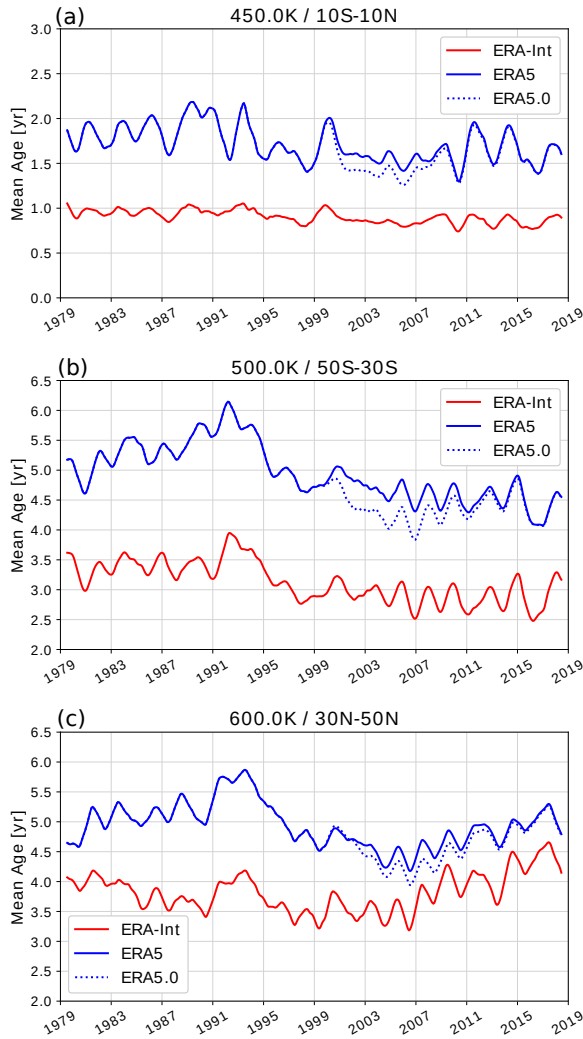

**Figure 9.** Mean age time series from ERA5 and ERA–Interim in the (a) tropics at 450 K and 10°S–10°N, in the (b) SH subtropics at 500 K and 30–50°S, and in the (c) SH subtropics at 600 K and 30–50°N. The blue dotted line shows mean age from the sensitivity model simulation driven with the uncorrected ERA5.0 data (see text). Data has been deseasonalized by applying a 12 month running mean.

of the uncertainty range of the in-situ $SF_6$-based observations. Recently, Leedham Elvidge et al. (2018) have shown that $SF_6$ based mean age may be biased high even outside of the polar vortex. On the other hand, the ERA5 age data shows a steeper latitudinal gradient in the subtropics which agrees slightly better with the steep gradient in in-situ observed age compared to ERA–Interim.

Comparison to the NH balloon-borne mean age data set of Engel et al. (2017) shows again that ERA5 age is at the upper margin of the observational uncertainty range before about 1997, while ERA–Interim is at the lower margin (Fig. 10b). Re-





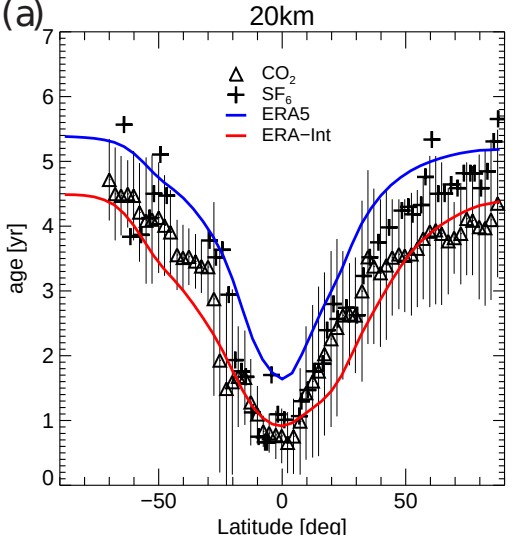

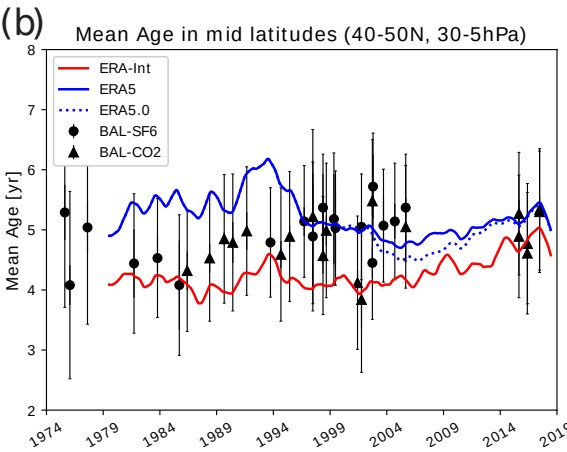

**Figure 10.** (a) Mean age of air at 20 km at different latitudes from in situ observations (black symbols, from Waugh, 2009), from ERA5 (blue) and ERA–Interim (red) driven CLaMS simulations. (b) Mean age time series in NH middle latitudes (40–50°N and 30–5 hPa, approximately 600–1200 K for reanalyses). Coloured lines show the mean age from reanalyses (smoothed with a 12–month running mean), black symbols show the mean age from the balloon observations of Engel et al. (2017), with error bars representing the uncertainty of the observations.

garding the trend, the more gradual increase in ERA–Interim mean age appears to compare better to the observed data than the temporal evolution of ERA5 mean age. In particular the strong decrease in ERA5 age in the mid-nineties is not present in the observations. Although computing a linear trend for a time series with a step change (like for ERA5) is questionable,

for completeness we note the trend values from a simple linear regression for ERA5 ($-0.13 \pm 0.01$ years/decade) and for ERA–Interim ($0.15 \pm 0.01$ years/decade). The trend value for ERA–Interim agrees with the value stated by Engel et al. (2017), although the observational trend is non-significant. Furthermore, Fritsch et al. (2020) showed recently that the calculated value





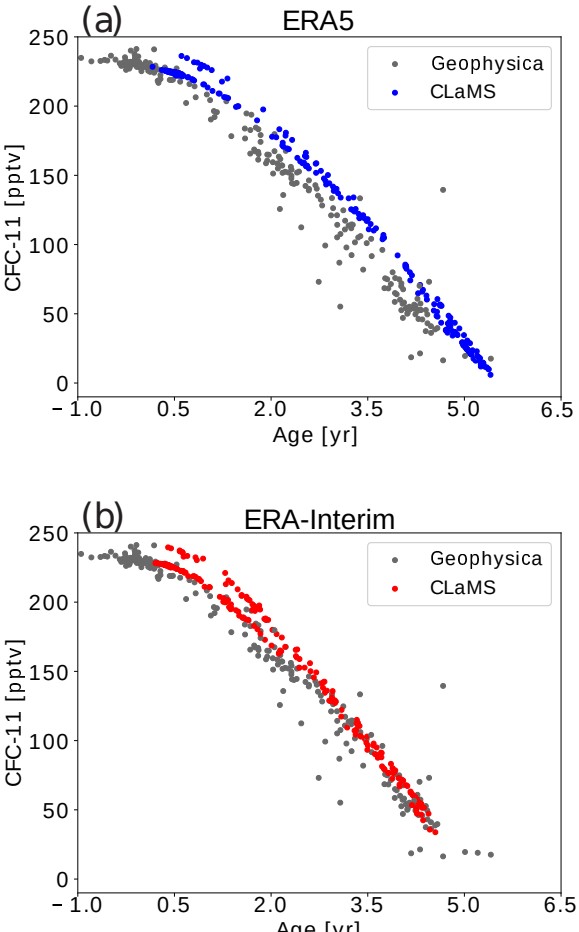

**Figure 11.** Correlation between CFC–11 and mean age from Geophysica aircraft observations (grey) and from (a) ERA5 (blue) and (b) ERA–Interim (red). Observations are from measurement campaigns during 2009–2017 covering the NH lower stratosphere (see text for details).

for the observed mean age trend depends critically on the estimated age spectrum shape, and approaches zero for more recent parameter settings suggested by (Hauck et al., 2019). Hence, the comparison between the ERA–Interim and the observed trend

value should not be over-interpreted, also given the differences between the single data points of the two time series in Fig. 10b.

Figure 11 further compares the reanalysis to in-situ observations from Geophysica high-altitude aircraft flights in the NH lower stratosphere (for a detailed measurement data description, see Laube et al., 2020). The comparison is done in terms of correlation between CFC–11 and mean age, as similar correlations between mean age and long-lived tracers have been

proven beneficial for identifying model transport deficits (e.g., Strahan et al., 2011). All data sets show the expected negative





correlation with lower CFC–11 mixing ratios related to larger age, because of stronger photochemical depletion in older air at upper stratospheric levels. Closer inspection shows that the ERA5 correlation is slightly shifted to the convex side of the observed correlation. This shift can be explained by the slower BDC in ERA5 compared to ERA–Interim, with air of the same age reaching not as deep into the stratosphere in ERA5 and hence experiencing less CFC–11 loss. A qualitatively similar, even

stronger, correlation shift was found for MERRA–2 reanalysis by Laube et al. (2020). For the faster BDC in ERA–Interim the CFC–11 versus age correlation agrees somewhat better with the in-situ observations.

## 6 Discussion

Recent studies have stated a substantial uncertainty in the climatological strength and trends of the stratospheric BDC and age of air in current generation reanalyses (e.g., Abalos et al., 2015; Miyazaki et al., 2016; Chabrillat et al., 2018; Ploeger et al.,

2019). Among the considered reanalyses, JRA–55 was shown to have the fastest and MERRA–2 the slowest BDC, with ERA–Interim-based results in between. In the present study, we find the BDC in ERA5 to be significantly slower and age of air significantly higher than in ERA–Interim (e.g., Figs. 1 and 3).

It is so far unclear whether the representation of the BDC in ERA5 is improved compared to the older reanalyses. In the tropical lower stratosphere, the weaker heating rates in ERA5 are consistent with slower residual circulation upwelling found

by Diallo et al. (2020) and appear to correct the 30–40% high bias in ERA–Interim heating rates in the tropical tropopause layer (TTL) found in previous studies (e.g., Ploeger et al., 2012; Schoeberl et al., 2012). On the other hand, the minimum in tropical upwelling around the level of zero radiative heating (around 350 K) is much lower for ERA5 than for ERA–Interim with heating rates even showing a gap in annual mean upwelling in a shallow layer, similar to the case in MERRA–2 reanalysis (e.g., compare Fig. 1 with Fig. 5 of Ploeger et al., 2019). In a recent paper, Wright et al. (2020) linked differences in reanalysis

diabatic heating rates to differences in the representation of clouds. They argued that the upwelling gap in ERA5 and MERRA–2 is likely caused by higher cloud water content in these two reanalyses and related radiative effects. Indeed, seasonal means of heating rates show upwelling also at the lower TTL levels, but very confined regionally and much weaker than for ERA–Interim. Hence, while in the tropical lower stratosphere the representation of tropical upwelling seems to be improved in ERA5, it is unclear whether the very weak total diabatic heating rates in the lower TTL are realistic.

At higher levels in the stratosphere, however, age of air in ERA5 is slightly high-biased compared to in-situ observations (Fig. 10a). In the lower stratosphere outside the tropics, a shift in the ERA5 CFC–11/age correlation with respect to aircraft and balloon in-situ observations further indicates that the BDC in ERA5 is somewhat too slow compared to observations (Fig. 11). It should be noted that these differences are small and that from our analysis the slow BDC in ERA5 is not necessarily incompatible with mean age observations, when taking all uncertainties into account. However, the ERA5 BDC is clearly at

the upper margin of the observational uncertainty range.

Comparison of age of air trends shows a similar hemispheric dipole pattern over the 2002–2012 period for ERA5 as for ERA–Interim, which was argued by Stiller et al. (2017) to agree qualitatively with the structural circulation change observed by MIPAS. This observed increase of age in the NH and decrease of age in the SH has so far not been found for "non-ECMWF"





reanalyses (Chabrillat et al., 2018; Ploeger et al., 2019). Regarding the long-term trend in the NH, ERA5 and ERA–Interim

show differences. While ERA–Interim shows weakly increasing age of air, more consistent with non-significant trend vaues
from balloon-borne in-situ observations (Engel et al., 2017; Fritsch et al., 2020), ERA5 shows decreasing age. Hence, overall
the agreement to stratospheric age of air observations appears slightly better for ERA–Interim compared to ERA5.

The globally negative age of air trend in ERA5 over 1989–2018 agrees with results from climate model simulations, show-
ing an accelerating BDC and decreasing age over multi-decadal time scales in response to increasing greenhouse gas con-

centrations. In the tropical lower stratosphere, the residual circulation upwelling increase of 2.4 %/dec in ERA5, as inferred
from RCTTs, agrees even quantitatively with climate model predictions of 2–3 %/dec (e.g., Butchart, 2014). In the lower trop-
ical stratosphere the residual circulation acceleration in ERA–Interim is similar (see Sect. 4). However, in ERA5 this residual
circulation acceleration reaches substantially higher whereas in ERA–Interim the deep BDC branch decelerates (Fig. 8).

In a recent study, Diallo et al. (2020) compared the residual circulation in ERA5 and ERA–Interim based on residual circu-

lation vertical velocity $\overline{w}^*$ and stream function. They showed, using these standard circulation metrics, that the BDC in ERA5
is significantly slower than in ERA–Interim, consistent with the findings here based on age of air and the diabatic circulation
(heating rate based). Furthermore, they related the weaker residual circulation to weaker gravity wave drag at the upper flanks
of the subtropical jets in ERA5 compared to ERA–Interim. Also differences in trends in $\overline{w}^*$ were shown to be likely caused by
differences in gravity wave drag. Given the qualitatively similar results between age of air and residual circulation regarding

both the weaker climatological BDC and more negative trends in ERA5, it seems likely that reanalysis differences between
ERA5 and ERA–Interim in age of air are mainly caused by the differences in residual circulation and that mixing differences
play only an amplifying role (as suggested for different data by Garny et al., 2014). The clear differences in the age spectrum
peaks (modal ages, compare Figs. 4, 5, 7) corroborate the idea that residual circulation differences cause the mean age dif-
ferences. An investigation of mixing in ERA5 based on computation of effective diffusivity (e.g., Haynes and Shuckburgh,

2000), as has been realized for other reanalyses recently (e.g., Abalos et al., 2017), could shed more light on the role of mixing
processes for age of air differences between the reanalyses.

A closer look shows that the decrease of mean age in ERA5 is not simply linear, particularly in the NH. Mean age time
series show even weak increases before about 1991 and after about 2000 and a decrease in between. Although these steplike
changes are also visible in ERA–Interim, to some degree, they are much more distinct in ERA5. Whether these changes in

mean age can be explained by known factors of multi-annual to decadal variabily or are related to changes in the reanalysis
assimilation system needs to be shown in future studies. If related to real variability, these decadal variations strongly overly the
potential long-term trend in ERA5 in the extratropical stratosphere. The 30-40 years time series is presumably still too short for
computing long-term trends in reanalysis, where stratospheric variability might be larger compared to climate models. Whether
the long-term mean age trend in ERA5 reanalysis is indeed negative needs to be further analyzed in the future by including

more years after 2018, and also before 1979 (as available from an extended ERA5 data product to be released soon).

As mentioned already in Sect. 2.2, we carried out CLaMS model simulations also with the preceding data set with an existing
bias in stratospheric temperatures (termed ERA5.0 here). Mean age time series from this sensitivity simulation are included
in Figs. 9 and 10 (dotted blue lines). The comparison between ERA5 and ERA5.0 shows that the bias correction indeed has a



significant effect on the stratospheric BDC. Without the correction, mean age values suddenly decrease around the year 2000
and remain lower during the following years compared to the corrected ERA5 data. Therefore, in ERA5.0 the steplike age
change is even stronger than in the corrected data. Consequently, trends over periods beginning during 2000–2010 (depending
on the region under consideration) are more strongly positive for ERA5.0 compared to the corrected ERA5 data. Furthermore,
the reduction of the steplike change in age due to the temperature bias correction raises the question whether the remaining
steplike change could be related to an incomplete bias correction in ERA5.

As explained in Sect. 2 the CLaMS model simulations are based on the diabatic circulation in the reanalysis as driven by the
reanalysis diabatic heating rate. For ERA–Interim it has been shown that the choice of a diabatic versus kinematic transport
representation could indeed change the simulated BDC to some degree, in particular regarding trends over decadal periods
(Chabrillat et al., 2018; Ploeger et al., 2019). Hence, comparison of the diabatic results here to a similar model study focussing
on the ERA5 BDC using a kinematic transport model would be particularly interesting. Furthermore, the present study is based
on ERA5 data with full vertical but truncated $1 \times 1$ degree horizontal resolution, as provided by ECMWF (see Sect. 2.2). We
assume that differences in the global scale BDC patterns caused by the truncation of horizontal resolution, and hence small-
scale mixing processes, will be minor. However, at this stage this is just an assumption and can not be proved as full CLaMS
simulations with full ERA5 resolution over the entire ERA5 period are so far not feasible due to the excessive amount of
reanalysis data needed and necessary further model developments regarding the data handling.

## 415  7   Conclusions

We investigated the global stratospheric Brewer-Dobson circulation in the ECMWF ERA5 reanalysis based on age of air
simulations with the Lagrangian chemistry transport model CLaMS driven by reanalysis winds and diabatic heating rates.
The simulations include both mean age as well as the age of air spectrum. Results are compared against results based on the
predecessor reanalysis ERA–Interim.

We find that the global structure and seasonality in both reanalyses is very similar. However, the BDC is substantially slower
and age of air larger in ERA5 than in ERA–Interim. In the tropical lower stratosphere and in the TTL the 30-40% weaker
heating rates in ERA5 appear to correct the too-strong vertical upwelling in ERA–Interim found in previous studies. At higher
stratospheric levels, on the other hand, ERA5 mean age is found to be at the upper margin of the observational uncertainty
range.

The mean age trend over the 1989–2018 period in ERA5 is globally negative, as expected from climate model simulations as
response to increasing greenhouse gas concentrations. However, outside the tropics the ERA5 mean age decrease is not linear
over the entire period but largely related to a steplike change around the year 2000. Hence, it is unclear whether the negative
age trend in the reanalysis can be interpreted as response to climate change or is related to decadal variability or changes in
the data assimilation system. Due to the slowness of the circulation and the existence of a negative age trend in the NH the
discrepancy with balloon-borne observations is greater for ERA5 than ERA–Interim. The mean age change over 2002–2012 in
ERA5 shows a similar hemispheric asymmetry pattern as found for ERA–Interim.



Overall, the new ERA5 reanalysis appears promising for transport studies of the BDC. It is certainly important for such studies to use the bias-corrected data set (termed ERA5.1 in ECMWF's documentation). However, whether the representation of stratospheric transport is indeed improved compared to ERA–Interim reanalysis is, so far, unclear.

*Data availability.* ERA5 and ERA–Interim reanalysis data is available from the ECMWF. The CLaMS model data used for this paper may be requested from the corresponding author (f.ploeger@fz-juelich.de)

*Author contributions.* FP carried out the CLaMS simulations and the respective analysis, with help from PK, and wrote the manuscript. MD, JUG and GG downloaded and prepared the ERA5 reanalysis data, with advice from BL. EC contributed to the analysis of the model output. JL and AE prepared and provided observational data. All authors contributed to finalizing the manuscript.

*Competing interests.* The authors declare no competing interests.

*Acknowledgements.* We are particularly grateful to Nicole Thomas for programming support and help with setting up the CLaMS model simulations with the new ERA5 reanalysis. We further thank Rolf Müller for comments on an earlier version of the manuscript, and the ECMWF for providing reanalysis data. This study was funded by the Helmholtz Association under grant no. VH-NG-1128 (Helmholtz Young Investigators Group A–SPECi). Finally, we gratefully acknowledge the computing time for the EMAC simulations which was granted
on the supercomputer JURECA at the Jülich Supercomputing Centre (JSC) under the VSR project ID JICG11.



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
