# Peer review of "The stratospheric Brewer–Dobson circulation inferred from age of air in the ERA5 reanalysis"

_Atmospheric Chemistry and Physics, 2020_

## Referee Comment (RC2)

**The stratospheric Brewer–Dobson circulation inferred from age ofair in the ERA5 reanalysis**

submitted to ACP by F. Ploeger et al.

S. Chabrillat, BIRA-IASB, February 2021

**General Comments**

The representation of the BDC and its time variations in reanalyses is an important topic for modellers of atmospheric composition in the middle atmosphere, and ERA-5 is set to become as broadly used as ERA-Interim has been for the twelve last years. Hence this study is of high interest and very timely for ACP. The modelling tools are very well tailored for these aims, and the modelling experiments are well designed. The manuscript is well structured and well written. All my comments deal with the description of the methods and observations, and the interpretation of the results. Hence I recommend publication of the paper after a minor revision (i.e. no additional calculations should be necessary).

In my opinion the manuscript comes short in the comparisons with trace gas observations (section 5). These suggest that the diabatic Age of Air (AoA) is overestimated in ERA-5, at least in the lower stratosphere at most latitudes (for some unspecified time period; Fig. 10a) and also in the middle-latitudes during the first half of the 1990's (Fig. 10b). It is repeatedly stated that the ERA-5 results are "*at the upper margin of the observational uncertainty range*" but these observational uncertainties are not clearly described in Figures 10 and 11, preventing a serious assessment of the significance of the apparent high bias using ERA-5. The AoA overestimation is confirmed by correlations between AoA and CFC-11, using as reference a campaign of aircraft observations (Fig. 11) but here also there is no indication of the observational uncertainties. These comparisons are briefly discussed in section 6 (lines 355-360) but mainly to tone down their importance and to avoid overinterpretation. The authors seem so unsure about these comparisons that they completely overlooked section 5 in the abstract.

It is well understood that these comparisons are not certain and that the same team is preparing a deeper investigation. Yet the results already shown in this manuscript are clearly of high interest for all readers and should be treated with due care. I recommend to better explain the observational uncertainties (see specific comments SC25 and SC26 below) and spend more effort to assess the significance of the overestimations of AoA by ERA-5 (see specific comment SC29 below). The abstract should highlight the results of section 5, and of course mention the discussion about their significance.

**Major comments**

**MC1.** The introduction and the discussion draw an often-stated parallel between the decreasing AoA found by some reanalyses in some hemispheres during some periods (maximum 30 years) and the decreasing AoA robustly found by most climate models over ~100 years due to increasing greenhouse gases. The validity of this parallel is doubtful, as correctly noted in the discussion (lines 392-393): "*The 30-40 years time series is presumably still too short for computing long-term trends in reanalysis, where stratospheric variability might be larger compared to climate models.*"
Fortunately this parallel is not necessary because AoA variations in the "actual world" over timescales of 1 to 3 decades are worth studying on their own. On this topic see also SC24 and SC31 below.

**MC2.** Section 2.1 explains two changes in methodology with respect to the previous age of air studies by this team: the mean age is now computed from a separate clock-tracer (rather than as the first moment of the AoA spectra); and the vertical velocity is not corrected any more for missing balance in the cross-isentropic mass flux. While the impacts of these changes are briefly described in that section and mentioned in section 4, they are not shown in the paper. In view of the long and successful series of AoA studies relying on ClaMS, it is important to show the impact of these changes on mean AoA. The impact of the first change especially could easily be shown on some existing figures, e.g. adding dashed black lines on Fig. 4 and 5 to show mean AoA from clock-tracer calculations.
It would be even more interesting to add a figure showing the Age spectrum for a given latitude and potential temperature. Ploeger et al. (2019) did precisely this in their Fig. 12a, showing clearly the importance of the spectrum tail in MERRA-2. In view of the apparent similarities between ERA-5 and MERRA-2 (when compared with ERA-I), this could be highly relevant here as well. Of course the mean AoA from the spectrum would also be compared with the mean AoA from the clock tracer.

**MC3.** Section 2 lacks a (short) description of the linear regression approach used for section 4: when de-seasonalized time series are regressed, do the linear regressions use "climate proxies" such as QBO, ENSO, volcanic forcings? If yes, what are the proxies actually used? The regression method also provides standard deviations that are used to assess the significance of the regressed trends (fig. 6). While this is standard, it remains important to explain how these standard deviations are ovtained (in a few words and with a reference).

**MC4.** Section 2.2 states (lines 139-140): "*However, we maintain the full vertical resolution. Hence, the ERA5 data to drive the ClaMS model in this study has 137 hybrid ECMWF model levels, **to compare with the** ERA–Interim 60 levels.*" How is it possible to maintain vertical resolution while going from sigma-pressure to sigma-theta? What is the actual vertical resolution in both simulations? The reader must get a sense that the ERA5 simulation makes full use of its increased vertical resolution. This is important as it certainly plays a role in the "diabatic heating gap" that is found at 350K (Fig. 1e) and repeatedly mentioned in the manuscript.

**MC5.** According to Diallo et al. (2020), the vertical coordinate that is used for derivation of the residual vertical velocity $\overline{w}^*$ is log-pressure height, i.e. a constant-pressure grid assuming a scale height of 7 km. Yet Fig. 2 uses "Altitude" for the Y-axis. This is confusing and prevents matching the levels of Fig. 2 with the iso-pressure levels explicitly drawn of Fig. 1, 3, 6, 8. If the actual Y-axis for Fig. 2 is log-pressure height, I recommend to re-draw Fig. 2 with pressures on the (logarithmic) Y-axes.

**MC6.** Fig. 8 compares the RCTT in ERA-5 and ERA-I, and the accompanying text states that "*Differences between mean age and RCTT are related to mixing effects*". Yet the corresponding paragraph discusses only the changes in RCTT and does not attempt to disentangle the contributions of mixing and residual circulation to AoA and its time variations. In practice it is not possible to visually compare AoA (Fig. 3) and RCTT (Fig.8) because AoA is shown sparately for DJF and JJA while RCTT is shown as an annual mean (also the color maps differ).

**Specific Comments**

**SC1.** Line 10: "*…changes in the assimilation system*" : reanalyses avoid any change in the assimilation system (i.e. the model and assimilation software); they suffer instead from changes in the observing system that is assimilated into the reanalysis. This formulation should be corrected throughout the manuscript.

**SC2.** Lines 16 and 20 : "*The Brewer-Dobson circulation (BDC) is the global transport circulation in the stratosphere*" and "*The BDC is characterized by upwelling motion in the tropics, poleward transport in the stratosphere and downwelling above middle and high latitudes.*" While these characterizations are very common, they overlook the important branch of the BDC that extends into the mesosphere where it goes all the way from the summer pole to the winter pole (e.g. Fig.1 in Bönisch et al., 2011).

**SC3.** Lines 40-43: "*On the one hand, climate models  predict a robust strengthening and acceleration of the BDC with climate change (e.g., Butchart et al., 2010), manifesting in an increase in tropical upwelling and a decrease in global mean age of air. On the other hand…*" On what timescale does the climate model prediction appear? See MC1: the opposition between long-term AoA changes in climate models and multi-decadal AoA changes in reanalyses is misleading and not necessary.

**SC4.** Line 53: "*…while inter-annual BDC variability seems to be well represented in reanalyses…*". This statement should be more specific and have specific references because it is really not obvious to me, e.g. Chabrillat et al. (2018, Fig.10) found that the amplitude of the QBO impact on AoA can vary by a factor of 2 depending on the input reanalysis.

**SC5.** First paragraph of page 3: consider inserting here a reference to Fujiwara et al. (2017) as it is a general introduction of the reanalyses and their diversity, with a comprehensive description of the observing systems that they assimilate.

**SC6.** Lines 91-94: this paragraph should be expanded to introduce more smoothly diabatic transport and its advantages in the stratosphere (with references). It is especially important to define $\theta$ before it is used.

**SC7.** Line 131: please provide a web link or reference to the ECMWF technical documentation for IFS CY41R2. "*The horizontal resolution is about 30km (T639)*": explain (in a few words) the spherical harmonics number.

**SC8.** Line 133: Simmons et al. (2020) describe the need for the ERA5.1 update and its differences with ERA5.0. It is really necessary to cite that reference here.

**SC9.** Line 138: what is the spectral truncature (wavenumber) corresponding to this 1°x1° resolution? This is important to compare with the T639 number provided above.

**SC10.** Lines 141-146: For clarity, it is necessary to first explain in a few words what is the "temperature tendency". Is the total diabatic heating rate $Q_{tot}$ the same quantity as the 5th temperature tendency provided by ERA, i.e. the "*mean temperature tendency due to* [both? all?] *parameterizations*"? Is $Q_{tot}$ nothing else than the sum of the temperature tendencies due to short-wave and long-wave parameterizations in all-sky conditions? Please clarify.

**SC11.** $d\theta/dt$ is variously described in the text as the "cross-isentropic vertical velocity" (e.g. lines 145 and 163) or as the "total diabatic heating rate" (e.g. caption of Fig. 1). In view of Eq (3), I think that the second formulation is not rigorous. In any case the same formulation should be used consistently throughout the manuscript.

**SC12.** Line 166: "*Cross-isentropic tropical upwelling maximizes in boreal winter*". But the equinox seasons are not shown or discussed anywhere, and it is problematic to write about tropical upwelling "in" boreal winter. Consider instead: "*Cross-isentropic tropical upwelling **is larger in DJF than in JJA.***" This comment applies throughout the manuscript.

**SC13.** Line 170: "*…at lowest TTL levels around the level of zero radiative heating **in ERA-5** (about 350K)...*" This is important since there is no such level in ERA-I. But maybe I did not understand correctly (see SC27 below) – in such case the text should be clarified.

**SC14.** Line 175: for clarity I suggest to insert here a preview of the Discussion, inserting few words e.g. "*This much weaker upwelling in the TTL and tropical lower stratosphere causes stronger restrictions on large-scale advective upward transport in ERA5 **and appears to correct an overestimation in ERA-I (see section 6).***"

**SC15.** Lines 178-179: "*The upward velocities in ERA5 in that region are more consistent with the residual circulation vertical velocity (see next paragraph and Fig. 2).*" This transition does not work – "see next paragraph" should be avoided. I suggest to move that sentence to the end of the next paragraph.

**SC16.** Lines 188 and 189: upwelling and downwelling are "stronger" but not "strongest" and tropical upwelling is stronger in DJF not in boreal winter (see SC12).

**SC17.** Lines 196-197: "*up to 30-40%, see Fig. 1*": normalized differences can not be seen on Fig.1 "*Above about 20 km...*": what is the corresponding theta? See MC5. A proper comparison would actually require a figure that overlays the two vertical profiles of tropical $d\theta/dt$ as function of $\theta$ with the two vertical profiles of tropical $\overline{w}^*$ as a function of $p$.

**SC18.** Figure 3c and 3d are difficult to read due to lack of contrast between the blue colors that are also obscured by the black contour lines. Consider removing these black color lines (pressure levels and zonal wind contours are well sufficient) and/or changing the color map.

**SC19.** Line 223: the differences are not so "*clear*" since closer inspection is required. In the tropics the differences are definitely not clearly visible and would require actual lineplots of the spectra (which could be worth adding; see MC2).

**SC20.** Line 226: the words "*against exchange with middle latitudes*" are superfluous

**SC21.** Line 237: "*…faster transport of young air towards the pole in ERA–Interim than in ERA5.*" This is not true everywhere: the northernmost latitudes in Fig. 5c show older modal age in ERA-I than in ERA-5.

**SC22.** Lines 244-252: the negative age trend with ERA-5 is significant in (nearly) the whole stratosphere for the period 1989-2018. Similarly, the trends found with ERA-I for 1989-2018 are either negative either positive but significant nearly everywhere. These significances should be highlighted.
Line 252: "*In the lowest tropical  stratosphere ERA5 shows **significantly** increasing age,  compared to decreasing age **or insignificant trends** in ERA–Interim in this region.*"

**SC23.** Lines 265-271: I find this paragraph quite confusing. Here is a suggestion for a clarified version (assuming I understood correctly):
"*The clearest difference to ERA–Interim regarding structural age spectrum changes emerges at middle and high latitudes at upper levels (here 600K, Fig. 7b and d). ERA5, on the one hand, shows a shift of the spectrum to younger ages ; ERA–Interim, on the other hand, shows a decrease in the fraction of air younger than about 4 years and an increase of the fraction of older air. This increas**ing** fraction of air older than about 4 years in ERA–Interim indicates **that ERA-5 has** a strengthening in the deep branch of the residual circulation. The different spectrum changes in ERA5 and ERA–Interim cause the opposite mean age changes in the two reanalyses (Fig. 6).*"
The end of this paragraph aims to introduce the follwing one: this transition should be re-worded (see also MC6).

**SC24.** Lines 299-304: please re-write for clarity and elaborate on the interpration. Here is a suggestion:
"*…mean age appears to increase before about 1991 **in both hemispheres** and after about the year 2000  **in the NH** and decreases in between. These steplike changes are evident in both reanalyses**. In the beginning of the 1990's they**  could be related to **the Pinatubo eruption (i.e.** true atmospheric variability) **while in the end of the 1990's they could be related** to changes in the **observing system assimilated by the** reanalyses. In particular…*"
Chabrillat et al. (2018) discussed the obscuring impact of the observing system change that happened less than a decade after Pinatubo (their section 3.2) so this reference may be used again here.

**SC25.** Figure 10a: It seems to me that these AoA use as source region the tropical 100 hPa levels (rather than the surface in all your other plots). Please check and (if correct) state this fact in the figure caption. What time period covered in the ClaMS simulations is used for this figure? Does it match the observational values taken from Waugh (2009)? What is the exact uncertainty range indicated by the error bars, i.e. do they show $1$-$\sigma$ (66% confidence level) or $2$-$\sigma$ (95% confidence level) on each side

of the observation? Better explanations about this figure will allow improving the corresponding discussion and go a long way in addressing the general comment.

**SC26.** Section 5 needs to be substantively improved (see General Comments). Here are some questions and superficial suggestions:

Lines 310-311: "*...ERA5 mean age values are just at **outside** the upper margin of the uncertainty range of the in-situ SF 6 -based observations. **This apparent overestimation of mean age in ERA5 may be even larger in reality because** Leedham Elvidge et al. (2018) have **recently** shown that…*"

Lines 315-316: "*..ERA5 age is at **or above** the upper margin of the observational uncertainty range before about 1997, while ERA–Interim is  in the lower  **part of the range** (Fig. 10b).*"

Line 317: "*...the more gradual increase in ERA–Interim mean age...*" : over what period?

Line 320: "*...we note the trend values from a simple linear regression…*": over what period?

Line 328 and caption of Fig.11: "*…(for a detailed measurement data description, see Laube et al., 2020).*" What tracer was used to derive this observational AoA? This should be stated in the text, along with an estimate of the associated uncertainty and also the uncertainty of the CFC-11 observations. It would be even better to show these uncertainties graphically in Fig. 11 (if possible).

**SC27.** Line 347: "*…the minimum in tropical upwelling around the level of zero radiative heating (around 350K) is much lower for ERA5 than for ERA–Interim...*". I do not understand this formulation: is there a "level of zero radiative heating" in ERA-I as well? Do you mean something else than the minimum value itself? This is related to SC13 above.

**SC28.** Lines 353-354: it would be good to remind the reader about the corresponding pressure ranges i.e. "*...while in the tropical lower stratosphere **(50-150hPa)** the representation of tropical upwelling seems to beimprovedin ERA5, it is unclear whether the very weak total diabatic heating rates in the lower TTL **(150-300hPa)** are realistic.*"

**SC29.** Lines 355-360: this paragraph should be revised (see general comment). More specifically:

Line 355: "*...age of air in ERA5 is slightly high-biased...*" – but according to Fig. 10a this bias is significant and still ~1 year older, i.e. as large as the difference between MERRA-2 and ERA-I in Chabrillat et al. (2018) or Ploeger et al. (2019). Hence "slightly" seems rather subjective to me…

Line 358: "*It should be noted that these differences are small...*" – with respect to what?

Line 359: "*…when taking all uncertainties into account.*" This is a quite vague formulation, especially considering that the uncertainties were not sufficiently described (see SC25 and SC26).

**SC30.** Line 362: "*...which was argued by Stiller et al. (2017) to agree qualitatively with the structural circulation change observed by MIPAS.*" These MIPAS observations of structural circulation changes were first reported by Stiller et al. (2012). The agreement between AoA trends in observations and in ERA-Interim was first reported by Mahieu et al. (2014) using observations of HCl.

**SC31.** Line 369: "*The globally negative age of air trend in ERA5 over 1989–2018 agrees with results from climate model simulations, showing an accelerating BDC and decreasing age over multi-decadal time scales in response to increasing greenhouse gas concentrations.*"
This agreement may very well be for different reasons than increasing GHG concentrations because in ERA-5 NH the decreasing age is entirely due to the decrease in the 1990's (~1993-2003) i.e. a timescale much shorter than in the CCM simulations (and probably related to Pinatubo eruption). See also MC1 and SC24 above.

**SC32.** Line 395: "*…also before 1979 (as available from an extended ERA5 data product to be released soon).*" We can expect this backward extension to bring its own artifacts as the BDC will be much less constrained by satellite observations, with results more akin to those by climate models. In such a case the differences between CCM and ERA5 would most probably be related to shortcomings in the underlying model of ERA5 (i.e. IFS is an NWP model and not yet a climate model).

**SC33.** Line 404: "*…raises the question whether the remaining steplike change could be related to an incomplete bias correction in ERA5.*"
Or maybe to a bias correction that started too late? A cursory look at the ECMWF report on ERA5.1 (Simmons et al., 2020) would help in this part of the discussion.

**SC34.** Line 421: "*...and in the TTL ...*" According to your own discussion, it is not clear that the upwelling changes in the TTL are a correction.

**SC35.** Line 427: "*...largely related to a steplike change around the year 2000.*"
See above: your figures rather show two steplinke changes, one around 1993 and the other one around 2000.

**SC36.** Line 435: what is the availability of theobservational data shown in Fig. 10 and 11?

**Typos, wording etc.**

The points below are only suggestions to improve the wording and also to highlight a few typos:
- Line 4: "*ERA5-based results are compared to those **using** the preceding ERA–Interim reanalysis.*"
- Line 34: Waugh and Hall (2002) is a review paper on this diagnostic, hence the "e.g." is not necessary
- Line 101: "*…a pulse **period** of 2 months*".
- Line 153: "*For instance, the forecast data **starting** at 6UTC and **on the** 5$^{th}$ hour forecast step…*". Same for line 161.
- Line 157: "***Here** the hourly ERA5 data is **not** used to drive model transport but data sub-sampled in time **is used instead, hence** the temperature tendencies…*"
- Line 178: "*where ERA5 shows upward **velocities** whereas **they are downward in** ERA–Interim.*"
- Line 205: "*…apparent in both reanalys**e**s.*"
- Line 207: "*...criti**c**ally…*"
- Line 220: abbreviation NH not yet defined
- Line 231: "*clear  differences occur in the **details of the** spectrum shape*"

- Line 246: "**However** above about 500 K **and in particular in the NH**, the signs of the trends…"
- Line 286: "*...shows increasing RCTTs,  **with strongest positive trends** in the NH.*"
- Line 290: "*…(compare **with** Fig.6).*"
- Line 293: "*…the  **periodicity** is similar at all locations...*" or maybe "*the **timing of the** variability is similar at alo locations*"
- Line 309: "***The latitudinal distribution in** ERA-Interim **(Fig. 10a)** agrees well with…*"
- Line 372: "*... this **accelaration of the** residual circulation ...*"

**Additional bibliographical references**

Bönisch, H. et al. :On the structural changes in the Brewer-Dobson circulation after 2000
*Atmos. Chem. Phys.,* **2011***, 11*, 3937-3948

Fujiwara, M. et al.: Introduction to the SPARC Reanalysis Intercomparison Project (S-RIP) and overview of the reanalysis systems. *Atmos. Chem. Phys.,* **2017***, 17*, 1417-1452

Mahieu, E. et al.: Recent Northern Hemisphere stratospheric HCl increase due to atmospheric circulation changes. *Nature,* **2014***, 515*, 104-107

Simmons, A. et al.: Global stratospheric temperature bias and other stratospheric aspects of ERA5 and ERA5.1. *ECMWF Tech. Mem.*, **2020**, doi: 10.21957/rcxqfmg0

---

## Author Comment (AC2)

**Reply to Reviewer 1**

We thank the Reviewer for the positive evaluation of the manuscript and the good comments. In the following, we address all comments and questions raised (Reviewer's comments in italics). Text changes in the manuscript are highlighted in color (except minor wording changes).

**General comments:**

*I think that the apparent step-change extratropical in age around the early-mid 1990s is an interesting result and one of the clearer differences between ERA5 and ERA-Interim. The fact that this does not appear clearly in observational data (e.g. Fig. 10) may bring into question the reliability of ERA5 trends. Its also interesting that this step-change is not apparent in the tropical upwelling (Diallo et al. 2020, their Fig 12). I might encourage the authors to elaborate a little more on this in the paper as the result is not very prominent. In particular, do any TEM diagnostics of the circulation show this step change, or is this only seen in age calculations? Is so, could the authors speculate as to why? If this step-change is not thought to be real, do the authors have any suggestions for what change in the assimilation scheme may be responsible?*

This is indeed a very good question - and not easy to answer. At first glance, the change in age of air in the mid-nineties appears as a sudden step-change, but a closer look shows that the change occurs over a few years, between about 1991-1995. This change is clearly evident in ERA5 age, and to a weaker degree also seen in ERA-Interim. The age time series in Fig. 9 show that the clearer step change in ERA5 in the mid-nineties is mainly a result of the positive age trend in the eighties and the age increase around 1991 which is likely related to the Pinatubo aerosol. The main difference to ERA-Interim is the trend over the eighties.

As suggested by the Reviewer, we further investigated basic meteorological variables for similar changes, and we considered both the residual circulation vertical velocity and the diabatic heating rate (new Fig. 12 in the revised manuscript). At upper stratospheric levels the heating rates show abrupt changes related to changes in the assimilation system (e.g., in 1998), as discussed for ERA-Interim e.g. by Abalos et al. (2015). However, in the lower stratosphere none of the two variables shows a step-like change in the mid-nineties which could be related to the age of air change. On the other hand, the heating rates show a decrease after the Pinatubo eruption (1991) in both reanalysis, related to the age increase during the same period. The difference in the strength of this effect between the two reanalysis, and also between heating rates and residual circulation velocity, are not clear to us.

We agree with the Reviewer that a more thorough discussion of these issues clearly improves the paper and we included a new figure (Fig. 12) showing the vertical velocity and heating rate changes and extended the discussion section 6 in this regard.

**Minor and Technical comments:**

L7: *Above: it wasnt clear to me what this was referring to as being above. Maybe in the mid-upper stratosphere would be clearer?*
Indeed, our wording here was not precise. We actually meant that ERA5 age appears somewhat high-biased outside the TTL at all locations where we compared to observations. However, we compared only at 20km (aircraft data, Fig. 10a), in NH middle latitudes above 24km (balloon data, Fig. 10b), and in the NH lower stratosphere between about 350-480K (aircraft and balloon data, Fig. 11). Hence, this statement should not be considered as too strong. To be more precise, we changed the wording to: "At 20 km and in the NH stratosphere, ERA5 age values are at the upper edge of the uncertainty range from tracer observations, indicating a comparatively slow BDC." Note that we also changed parts of the comparison to observations in Sect. 5, as suggested by Reviewer 2 (we replaced the age-F11 correlation analysis with an analysis of the age difference distributions between model and observations).

L94: *theta (potential temperature) should be defined.*
Sentence has been changed accordingly.

L105: *along → along with?*
We just deleted "along".

Fig. 2: *I think this would benefit if a difference plot were also shown (ERA-Int minus ERA5) to aid with comparison of the two reanalyses and with Fig 1 e-g. It is quite difficult to pick out the differences without such a plot.*
Thanks for this suggestion, which eases the comparison! We just added difference plots to all cases (ERA5 and ERA-Interim, DJF and JJA) in Figs. 1 and 2.

Fig. 3: *This is a minor point, but I would encourage the authors to consider using a perceptually uniform color scale for the age plots (a,b,d,e), such as grayscale, viridis etc. The rainbow scheme used here can introduce the appearance of false boundaries (where the yellow color are) in the date. The same goes for Figs 4 and 5.*
We agree that the used color scheme could probably be improved. However, most publications we are aware of show age of air usig a similar blue–green–red color scheme. As readers therefore are mostly used to that we would stay with it here.

L391: *strongly overly. Im not sure what this means? Perhaps decadal variations are significant compared to potential long-term trends?*
We changed the sentence as suggested.

L427: *steplike change around the year 2000: To me it looks like the main steplike change in ERA5 is over 1992-1997 rather than around 2000.*
This is totally correct, and the "2000" was just wrong - Thanks for pointing this out! We changed the text to "in the mid-nineties".

---

## Author Comment (AC3)

**Reply to Reviewer 2**

We thank the Reviewer Simon Chabrillat for the very careful review, the many detailed comments, and the generally positive evaluation of the manuscript. In the following, we address all comments and questions raised (Reviewer's comments in italics). Text changes in the manuscript are highlighted in color (except minor wording changes).

**General comments:**

*The representation of the BDC and its time variations in reanalyses is an important topic for modellers of atmospheric composition in the middle atmosphere, and ERA-5 is set to become as broadly used as ERA-Interim has been for the twelve last years. Hence this study is of high interest and very timely for ACP. The modelling tools are very well tailored for these aims, and the modelling experiments are well designed. The manuscript is well structured and well written. All my comments deal with the description of the methods and observations, and the interpretation of the results. Hence I recommend publication of the paper after a minor revision (i.e. no additional calculations should be necessary).*

*In my opinion the manuscript comes short in the comparisons with trace gas observations (section 5). These suggest that the diabatic Age of Air (AoA) is overestimated in ERA-5, at least in the lower stratosphere at most latitudes (for some unspecified time period; Fig. 10a) and also in the middle-latitudes during the first half of the 1990s (Fig. 10b). It is repeatedly stated that the ERA-5 results are at the upper margin of the observational uncertainty range but these observational uncertainties are not clearly described in Figures 10 and 11, preventing a serious assessment of the significance of the apparent high bias using ERA-5. The AoA overestimation is confirmed by correlations between AoA and CFC-11, using as reference a campaign of aircraft observations (Fig. 11) but here also there is no indication of the observational uncertainties. These comparisons are briefly discussed in section 6 (lines 355-360) but mainly to tone down their importance and to avoid overinterpretation. The authors seem so unsure about these comparisons that they completely overlooked section 5 in the abstract.*

*It is well understood that these comparisons are not certain and that the same team is preparing a deeper investigation. Yet the results already shown in this manuscript are clearly of high interest for all readers and should be treated with due care. I recommend to better explain the observational uncertainties (see specific comments SC25 and SC26 below) and spend more effort to assess the significance of the overestimations of AoA by ERA-5 (see specific comment SC29 below). The abstract should highlight the results of section 5, and of course mention the discussion about their significance.*

We agree that the submitted manuscript did not present a too detailed comparison between model and observations and that this part could be further improved. However, as the Reviewer already recognized, we can not give too strong conclusions due to the scarcity of available observational data and their uncertainty. In the revised manuscript, we follow the Reviewer's suggestions and included (1) a more detailed description and discussion of the observational uncertainties, and (2) a clearer comparison of model age of air to Geophysica in-situ observations, as explained in detail in the following:

(1) Section 5 now includes more details on the observations by Engel et al. (2017), Laube et al. (2020), and those compiled by Waugh and Hall (2002). As the Waugh and Hall (2002) data set represents a standard benchmark for model evaluation, and has been used by many other authors before (e.g., Diallo et al., 2012; Chabrillat et a., 2018; Ploeger et al., 2019) we just added a few more details, like the period (1992–1997) and data type (ER–2 aircraft observations), and refer clearly to the Waugh and Hall publication. Furthermore, we used this data set as a benchmark for the model climatology, although the data are only from the nineties. However, similar comparisons to model climatologies have been made in many other studies (e.g., Diallo et al., 2012; Chabrillat et a., 2018; Ploeger et al., 2019). We included a short note on these issues in Sect. 5.
Also the data by Engel et al. (2017) have been well described in the literature (Engel et al., 2009, 2017). We mainly added a short description on the uncertainty, which is related to the sampling (representativeness), tropospheric mixing ratios, the used age spectrum parameterization and the absolute measurement error.
The data by Laube et al. (2020) is the newest data set and we added more details here, although also these data are already well described in the above mentioned paper. We now emphasize that these data are from 5 aircraft measurement campaigns with the Geophysica high-altitude research aircraft during 2009–2017. In particular, we added an analysis and description of the observational uncertainty, leading to an overall uncertainty range of $\pm 0.78$ years (see Sect. 5).

(2) We agree that the comparison in terms of correlations was somewhat complicated and not clear regarding the uncertainty ranges. We entirely changed this part of the paper. Instead of the correlations, in the revised version we just compare mean age of air. The new Fig. 11 presents the mean age model bias in terms of distributions of the differences model minus observations. The estimated overall uncertainty range for the observations (see point 1, above) allows assessing what fraction of data points are outside this uncertainty range. The fraction outside the uncertainty range is 21% for ERA–Interim and 56% for ERA5. Hence, ERA5 indeed appears somewhat high-biased. However, the 1-$\sigma$ ranges of the mean model bias and the observational uncertainty overlap (see Fig. 11), such that the bias is not significant at 66% confidence level. The related discussion in Sect. 5 and in the discussion is entirely rewritten and improved. Thanks for this very good suggestion which clearly improves and clarifies the paper!

**Major comments:**

MC1. *The introduction and the discussion draw an often-stated parallel between the decreasing AoA found by some reanalyses in some hemispheres during some periods (maximum 30 years) and the decreasing AoA robustly found by most climate models over 100 years due to increasing greenhouse gases. The validity of this parallel is doubtful, as correctly noted in the discussion (lines 392-393): The 30-40 years time series is presumably still too short for computing long-term trends in reanalysis, where stratospheric variability might be larger compared to climate models. Fortunately this parallel is not necessary because AoA variations in the actual world over timescales of 1 to 3 decades are worth studying on their own. On this topic see also SC24 and SC31 below.*
We fully agree with the Reviewer that studying AoA variations over shorter than centennial time scales is interesting and important. But it is also of strong interest whether the climate model predicted trend can be seen already in the reanalysis over the last decades. We added some explanatory text in the discussion Sect. 6.

MC2. *Section 2.1 explains two changes in methodology with respect to the previous age of air studies by this team: the mean age is now computed from a separate clock-tracer (rather than as the first moment of the AoA spectra); and the vertical velocity is not corrected any more for missing balance in the cross-isentropic mass flux. While the impacts of these changes are briefly described in that section and mentioned in section 4, they are not shown in the paper. In view of the long and successful series of AoA studies relying on ClaMS, it is important to show the impact of these changes on mean AoA. The impact of the first change especially could easily be shown on some existing figures, e.g. adding dashed black lines on Fig. 4 and 5 to show mean AoA from clock-tracer calculations.*
*It would be even more interesting to add a figure showing the Age spectrum for a given latitude and potential temperature. Ploeger et al. (2019) did precisely this in their Fig. 12a, showing clearly the importance of the spectrum tail in MERRA-2. In view of the apparent similarities between ERA-5 and MERRA-2 (when compared with ERA-I), this could be highly relevant here as well. Of course the mean AoA from the spectrum would also be compared with the mean AoA from the clock tracer.*
We agree that it is important to quantify the uncertainties in the methodology. However, the difference between mean age calculated from the age spectrum and from a clock-tracer has been analysed in detail and presented for the same model set-up already by Ploeger and Birner (2016), their Figs. 4 and A1. We included a reference to this paper at the appropriate place in the methods section. Regarding the age spectrum comparison at a given latitude and potential temperature, we think that Figs. 4 and 5 already present exactly this information, even for all latitudes at 400 and 600 K. As the paper includes already 12 figures we refrain from adding an additional line plot, which would just double already existing information.

MC3. *Section 2 lacks a (short) description of the linear regression approach used for section 4: when de-seasonalized time series are regressed, do the linear regressions use climate proxies such as QBO, ENSO, volcanic forcings? If yes, what are the proxies actually used? The regression method also provides standard deviations that are used to assess the significance of the regressed trends (fig. 6). While this is standard, it remains important to explain how these standard deviations are ovtained (in a few words and with a reference).*
We used just a simple linear regression of deseasonalized time series (after subtracting the mean annual cycle), hence without subtraction of variability related to e.g. QBO or ENSO. This information was already provided in the submitted manuscript at the beginning of Sect. 4, where the trends are presented and discussed. We tried to further clarify the wording of this paragraph and added information on the significance of the calculated trends in the revised version.

MC4. *Section 2.2 states (lines 139-140): However, we maintain the full vertical resolution. Hence, the ERA5 data to drive the ClaMS model in this study has 137 hybrid ECMWF model levels, to compare with the ERAInterim 60 levels. How is it possible to maintain vertical resolution while going from sigma-pressure to sigma-theta? What is the actual vertical resolution in both simulations? The reader must get a sense that the ERA5 simulation makes full use of its increased vertical resolution. This is important as it certainly plays a role in the diabatic heating gap that is found at 350K (Fig. 1e) and repeatedly mentioned in the manuscript.*
As ClaMS is a Lagrangian transport model, the grid points are provided by air parcels which are irregularly distributed in space. Hence, meteorological data (e.g., winds) need to be interpolated onto the air parcel positions.

To maintain full reanalysis vertical resolution, this interpolation is done from fields on native model levels. In that sense, the input reanalysis data for CLaMS has the original reanalysis vertical resolution (137 levels for ERA5, 60 levels for ERA–Interim). We added information and tried to clarify the respective text in the methods section.

MC5. *According to Diallo et al. (2020), the vertical coordinate that is used for derivation of the residual vertical velocity $\overline{w}^*$ is log-pressure height, i.e. a constant-pressure grid assuming a scale height of 7 km. Yet Fig. 2 uses Altitude for the Y-axis. This is confusing and prevents matching the levels of Fig. 2 with the iso-pressure levels explicitly drawn of Fig. 1, 3, 6, 8. If the actual Y-axis for Fig. 2 is log-pressure height, I recommend to re-draw Fig. 2 with pressures on the (logarithmic) Y- axes.*
Sorry for not having been clear here! The vertical coordinate used for calculating residual circulation velocity here is indeed log-pressure altitude, as is the standard framework for Transformed Eulerian Mean computations. In the revised Fig. 2 we clearly state this in the caption.

MC6. *Fig. 8 compares the RCTT in ERA-5 and ERA-I, and the accompanying text states that Differences between mean age and RCTT are related to mixing effects. Yet the corresponding paragraph discusses only the changes in RCTT and does not attempt to disentangle the contributions of mixing and residual circulation to AoA and its time variations. In practice it is not possible to visually compare AoA (Fig. 3) and RCTT (Fig.8) because AoA is shown sparately for DJF and JJA while RCTT is shown as an annual mean (also the color maps differ).*
The goal with presenting the RCTTs is to discuss the pure residual circulation effect on stratospheric transit times. Indeed, these RCTTs can not be directly compared to mean age, therefore by purpose we chose different plotting styles and averaging periods. We added explanation and tried to clarify the paragraph explaining the differences between RCTTs and mean age in Sect. 4.

**Specific comments:**

SC1. *Line 10: ...changes in the assimilation system : reanalyses avoid any change in the assimilation system (i.e. the model and assimilation software); they suffer instead from changes in the observing system that is assimilated into the reanalysis. This formulation should be corrected throughout the manuscript.*
The formulation "assimilation system" was meant to also included the observational data. Nevertheless, to enhance clarity we changed the wording as suggested to "observations included in the assimilation system" throughout the manuscript.

SC2. *Lines 16 and 20 : The Brewer-Dobson circulation (BDC) is the global transport circulation in the stratosphere and The BDC is characterized by upwelling motion in the tropics, poleward transport in the stratosphere and downwelling above middle and high latitudes. While these characterizations are very common, they overlook the important branch of the BDC that extends into the mesosphere where it goes all the way from the summer pole to the winter pole (e.g. Fig.1, in Bnisch et al., 2011).*
A note on the mesospheric circulation branch is added.

SC3. *Lines 40-43: On the one hand, climate models  predict a robust strengthening and acceleration of the BDC with climate change (e.g., Butchart et al., 2010), manifesting in an increase in tropical upwelling and a decrease in global mean age of air. On the other hand... On what timescale does the climate model prediction appear? See MC1: the opposition between long-term AoA changes in climate models and multi-decadal AoA changes in reanalyses is misleading and not necessary.*
See our reply to MC1 above.

SC4. *Line 53: ...while inter-annual BDC variability seems to be well represented in reanalyses.... This statement should be more specific and have specific references because it is really not obvious to me, e.g. Chabrillat et al. (2018, Fig.10) found that the amplitude of the QBO impact on AoA can vary by a factor of 2 depending on the input reanalysis.*
The formulation has been specified, by removing the statement on "inter-annual variability" and focussing just on "decadal changes".

SC5. *First paragraph of page 3: consider inserting here a reference to Fujiwara et al. (2017) as it is a general introduction of the reanalyses and their diversity, with a comprehensive description of the observing systems that they assimilate.*
A short note on S-RIP and the Fujiwara et al. (2017) paper is added.

SC6. *Lines 91-94: this paragraph should be expanded to introduce more smoothly diabatic transport and its advantages in the stratosphere (with references). It is espeecially important to define θ before it is used.*
We expanded the respective paragraph a bit (e.g., properly introducing $\theta$, adding a reference), but not too

much, as we think the advantages of a diabatic transport formulation for the stratosphere are well-known. Too much discussion of these aspects here would rather distract the reader from the main points of the paper.

SC7. *Line 131: please provide a web link or reference to the ECMWF technical documentation for IFS CY41R2. The horizontal resolution is about 30km (T639): explain (in a few words) the spherical harmonics number.*
We just added another reference to the Hersbach et al. (2020) paper here, although it was already referenced at the end of the paragraph.

SC8. *Line 133: Simmons et al. (2020) describe the need for the ERA5.1 update and its differences with ERA5.0. It is really necessary to cite that reference here.*
Reference to Simmons et al. (2020) has been added.

SC9. *Line 138: what is the spectral truncature (wavenumber) corresponding to this 1x1 resolution? This is important to compare with the T639 number provided above.*
We did a transformation on-the-fly during the download as provided by the ECMWF MARS system, which is a direct transformation from T1279 to 1.0/1.0 degree, with an automatic truncation to T213. This is detailed now in the revised manuscript.

SC10. *Lines 141-146: For clarity, it is necessary to first explain in a few words what is the temperature tendency. Is the total diabatic heating rate Qtot the same quantity as the 5th temperature tendency provided by ERA, i.e. the mean temperature tendency due to [both? all?] parameterizations? Is Qtot nothing else than the sum of the temperature tendencies due to short-wave and long-wave parameterizations in all-sky conditions? Please clarify.*
Yes, the total diabatic heating rate (used here for driving CLaMS transport) is the 5th temperature tendency due to all parametrisations, including the mentioned radiative contributions, latent heat release, turbulent and sensitive heating. We added this information and clarified the text.

SC11. *$d\theta/dt$ is variously described in the text as the cross-isentropic vertical velocity (e.g. lines 145 and 163) or as the total diabatic heating rate (e.g. caption of Fig. 1). In view of Eq (3), I think that the second formulation is not rigorous. In any case the same formulation should be used consistently throughout the manuscript.*
Yes, the Reviewer is totally right with this comment that the vertical cross-isentropic velocity $d\theta/dt$ used for driving CLaMS transport is not exactly the reanalysis diabatic heating rate, but is calculated from the heating rate using Eq. 3. Actually, we already tried to use the wording carefully in the submitted manuscript, but as the comment shows with limited success. We went through the paper again and tried to further clarify the wording, such that when it is referred to the reanalysis quantity (or the driving quantity) we use "diabatic heating rate", when it is referred to the velocity used in CLaMS we use "diabatic (or cross-isentropic) vertical velocity". We hope this point is much clearer now.

SC12. *Line 166: Cross-isentropic tropical upwelling maximizes in boreal winter. But the equinox seasons are not shown or discussed anywhere, and it is problematic to write about tropical upwelling in boreal winter. Consider instead: Cross-isentropic tropical upwelling is larger in DJF than in JJA. This comment applies throughout the manuscript.*
We think it is frequently used wording in the literature to link "tropical upwelling maximum" to "boreal winter", as the upwelling seasonality is controlled by the hemispheric asymmetry in wave driving (stronger in the NH). Hence, we leave the formulations largely as they are, but checked again for consistency, and also included a short addition at first usage "boreal winter (December–February, DJF)".

SC13. *Line 170: ...at lowest TTL levels around the level of zero radiative heating in ERA-5 (about 350K)... This is important since there is no such level in ERA-I. But maybe I did not understand correctly (see SC27 below) in such case the text should be clarified.*
The text here refers to the general location of the level of zero radiative heating, not to a specific ERA5 level. We slightly modified the text to "in the lowest TTL around the level of zero radiative heating" to avoid mentioning specific TTL levels, and hope this is clearer now.

SC14. *Line 175: for clarity I suggest to insert here a preview of the Discussion, inserting few words e.g. This much weaker upwelling in the TTL and tropical lower stratosphere causes stronger restrictions on large-scale advective upward transport in ERA5 and appears to correct an overestimation in ERA-I (see section 6).*
Done.

SC15. *Lines 178-179: The upward velocities in ERA5 in that region are more consistent with the residual circulation vertical velocity (see next paragraph and Fig. 2). This transition does not work see next paragraph should be avoided. I*

*suggest to move that sentence to the end of the next paragraph.*
Wording has been modified.

SC16. *Lines 188 and 189: upwelling and downwelling are stronger but not strongest and tropical upwelling is stronger in DJF not in boreal winter (see SC12).*
Changed. However, we kept the wording "boreal winter" (see reply to SC12).

SC17. *Lines 196-197: up to 30-40%, see Fig. 1: normalized differences can not be seen on Fig.1 Above about 20 km...: what is the corresponding theta? See MC5. A proper comparison would actually require a figure that overlays the two vertical profiles of tropical dθ/dt as function of θ with the two vertical profiles of tropical $\overline{w}^*$ as a function of p.*
As $\overline{w}^*$ is the vertical velocity in log-pressure coordinates, while dθ/dt is the vertical velocity in isentropic coordinates, we think it is better to show each variable in the respective coordinate system. However, we added difference plots to all cases in Figs. 1 and 2 (as also suggested by Reviewer 1) to ease comparison and evaluation of the differences.

SC18. *Figure 3c and 3d are difficult to read due to lack of contrast between the blue colors that are also obscured by the black contour lines. Consider removing these black color lines (pressure levels and zonal wind contours are well sufficient) and/or changing the color map.*
We agree that the contrast in Fig. 3 is not optimal. However, we think it is better to keep the black contour lines of climatological mean age to ease evaluation of the relative differences. Also, we prefer to have a bipolar blue–red color map for difference plots, although the differences in that case are all negative.

SC19. *Line 223: the differences are not so clear since closer inspection is required. In the tropics the differences are definitely not clearly visible and would require actual lineplots of the spectra (which could be worth adding; see MC2).*
Although probably not optimal, we think the clarity of the differences is still good. As explained in the reply to MC2 we don't want to include more figures to have the paper not too lengthy. Hence, we just modified the wording here slightly, saying now "Closer inspection reveals differences...".

SC20. *Line 226: the words against exchange with middle latitudes are superfluous*
Changed accordingly.

SC21. *Line 237: ...faster transport of young air towards the pole in ERAInterim than in ERA5. This is not true everywhere: the northernmost latitudes in Fig. 5c show older modal age in ERA-I than in ERA-5.*
This statement concerned poleward transport in the summer hemisphere, as said at the beginning of the sentence. To make this clearer, we added a remark on the DJF case where poleward transport is faster in ERA–Interim in the SH (Fig. 5c). We hope this is clearer now.

SC22. *Lines 244-252: the negative age trend with ERA-5 is significant in (nearly) the whole stratosphere for the period 1989-2018. Similarly, the trends found with ERA-I for 1989-2018 are either negative either positive but significant nearly everywhere. These significances should be highlighted. Line 252: In the lowest tropical and sub-tropical stratosphere ERA5 shows significantly increasing age, although non-significant changes in some regions, compared to decreasing age or insignificant trends in ERAInterim in this region.*
Changed accordingly.

SC23. *Lines 265-271: I find this paragraph quite confusing. Here is a suggestion for a clarified version (assuming I understood correctly): The clearest difference to ERAInterim regarding structural age spectrum changes emerges at middle and high latitudes at upper levels (here 600K, Fig. 7b and d). ERA5, on the one hand, shows a shift of the spectrum to younger ages, although not as clear as in the tropics and in the SH. ; ERA Interim, on the other hand, shows a decrease in the fraction of air younger than about 4 years and an increase of the fraction of older air. This increasing fraction of air older than about 4 years in ERA Interim indicates that ERA-5 has a strengthening in the deep branch of the residual circulation. The different spectrum changes in ERA5 and ERAInterim cause the opposite mean age changes in the two reanalyses (Fig. 6)., and are related to different trends in the deep BDC branch (see next paragraph). The end of this paragraph aims to introduce the follwing one: this transition should be re-worded (see also MC6).*
We tried to further clarify this text, taking into account these comments and also a comment from Reviewer 1.

SC24. *Lines 299-304: please re-write for clarity and elaborate on the interpration. Here is a suggestion: ...mean age appears to increase before about 1991 in both hemispheres and after about the year 2000 (except in the SH) in the NH and decreases in between. These steplike changes are evident in both reanalyses. In the beginning of the 1990s they and could be related to the Pinatubo eruption (i.e. true atmospheric variability) while in the end of the 1990s they could be related*

*to changes in the observing system assimilated by the reanalyses. In particular... Chabrillat et al. (2018) discussed the obscuring impact of the observing system change that happened less than a decade after Pinatubo (their section 3.2) so this reference may be used again here.*

The text has been modified accordingly, and the Chabrillat et al. reference added here.

SC25. *Figure 10a: It seems to me that these AoA use as source region the tropical 100 hPa levels (rather than the surface in all your other plots). Please check and (if correct) state this fact in the figure caption. What time period covered in the ClaMS simulations is used for this figure? Does it match the observational values taken from Waugh (2009)? What is the exact uncertainty range indicated by the error bars, i.e. do they show 1-σ (66% confidence level) or 2-σ (95% confidence level) on each side of the observation? Better explanations about this figure will allow improving the corresponding discussion and go a long way in addressing the general comment.*

See mainly our reply to the general comment. Regarding the additional specific points here: The CLaMS age shown in Fig. 10a is consistent with the other data shown in the paper, hence with respect to the surface. The time periods in observational age and model age are, indeed, not the same (see general comment reply). This is stated now explicitly, with a more detailed description of observational uncertainties, in Sect. 5.

SC26. *SC26. Section 5 needs to be substantively improved (see General Comments). Here are some questions and superficial suggestions:*
*Lines 310-311: ...ERA5 mean age values are just at outside the upper margin of the uncertainty range of the in-situ $SF_6$ -based observations. This apparent overestimation of mean age in ERA5 may be even larger in reality because Leedham Elvidge et al. (2018) have recently shown that...*
*Lines 315-316: ..ERA5 age is at or above the upper margin of the observational uncertainty range before about 1997, while ERAInterim is at in the lower margin part of the range (Fig. 10b).*
*Line 317: ...the more gradual increase in ERAInterim mean age... : over what period?*
*Line 320: ...we note the trend values from a simple linear regression...: over what period?*
*Line 328 and caption of Fig.11: ...(for a detailed measurement data description, see Laube et al., 2020). What tracer was used to derive this observational AoA? This should be stated in the text, along with an estimate of the associated uncertainty and also the uncertainty of the CFC-11 observations. It would be even better to show these uncertainties graphically in Fig. 11 if possible).*

We modified and rewrote substantial parts of Sect. 5 as suggested and exchanged the correlation figure with a figure showing the difference between model and observation age (new Fig. 11), as already explained in the reply to the general comments above.

SC27. *Line 347: ...the minimum in tropical upwelling around the level of zero radiative heating (around 350K) is much lower for ERA5 than for ERAInterim.... I do not understand this formulation: is there a level of zero radiative heating in ERA-I as well? Do you mean something else than the minimum value itself? This is related to SC13 above.*

See our reply to SC13 above.

SC28. *Lines 353-354: it would be good to remind the reader about the corresponding pressure ranges i.e. ...while in the tropical lower stratosphere (50-150hPa) the representation of tropical upwelling seems to beimproved in ERA5, it is unclear whether the very weak total diabatic heating rates in the lower TTL (150-300hPa) are realistic.*

Approximate pressure ranges have been added to the text, as suggested.

SC29. *Lines 355-360: this paragraph should be revised (see general comment). More specifically: Line 355: ...age of air in ERA5 is slightly high-biased... but according to Fig. 10a this bias is significant and still 1 year older, i.e. as large as the difference between MERRA-2 and ERA-I in Chabrillat et al. (2018) or Ploeger et al. (2019). Hence slightly seems rather subjective to me... Line 358: It should be noted that these differences are small... with respect to what? Line 359: ...when taking all uncertainties into account. This is a quite vague formulation, especially considering that the uncertainties were not sufficiently described (see SC25 and SC26).*

This paragraph has been modified substantially, taking into account the Reviewer's suggestions and the improved comparison to in-situ observation-based age of air (see also reply to the general comment above).

SC30. *Line 362: ...which was argued by Stiller et al. (2017) to agree qualitatively with the structural circulation change observed by MIPAS. These MIPAS observations of structural circulation changes were first reported by Stiller et al. (2012). The agreement between AoA trends in observations and in ERA-Interim was first reported by Mahieu et al. (2014) using observations of HCl.*

The sentence here refers to the agreement between ERA–Interim and MIPAS age trends, not to the MIPAS age trends themselves. Therefore, we think the given reference to Stiller et al. (2017) is appropriate. The MIPAS trends are introduced already in the abstract, with proper referencing to the earlier paper by Stiller et al. (2012).

**SC31.** *Line 369: The globally negative age of air trend in ERA5 over 19892018 agrees with results from climate model simulations, showing an accelerating BDC and decreasing age over multi- decadal time scales in response to increasing greenhouse gas concentrations. This agreement may very well be for different reasons than increasing GHG concentrations because in ERA-5 NH the decreasing age is entirely due to the decrease in the 1990s ( 1993-2003) i.e. a timescale much shorter than in the CCM simulations (and probably related to Pinatubo eruption). See also MC1 and SC24 above.*
We agree that the age decrease in ERA5 is likely due to the age decrease in the mid-nineties. We discuss this aspect in the following paragraphs. We tried to further clarify the wording here and at other relevant places in the manuscript. (See also our replies to MC1 and SC24).

**SC32.** *Line 395: ...also before 1979 (as available from an extended ERA5 data product to be released soon). We can expect this backward extension to bring its own artifacts as the BDC will be much less constrained by satellite observations, with results more akin to those by climate models. In such a case the differences between CCM and ERA5 would most probably be related to shortcomings in the underlying model of ERA5 (i.e. IFS is an NWP model and not yet a climate model).*
This is a very valid remark, and we agree with this view. But still it will be very interesting to investigate the extended time period before 1979 in ERA5. Hence, we keep the formulation in the manuscript as is.

**SC33.** *Line 404: ...raises the question whether the remaining steplike change could be related to an incomplete bias correction in ERA5. Or maybe to a bias correction that started too late? A cursory look at the ECMWF report on ERA5.1 (Simmons et al., 2020) would help in this part of the discussion.*
We agree, and the suggested subsentence has been added.

**SC34.** *Line 421: ...and in the TTL ... According to your own discussion, it is not clear that the upwelling changes in the TTL are a correction.*
Thanks for pointing this inconsistency in our formulation out! We meant "upper TTL" here, and changed the text accordingly.

**SC35.** *Line 427: ...largely related to a steplike change around the year 2000. See above: your figures rather show two steplinke changes, one around 1993 and the other one around 2000.*
Again, thanks for pointing to that! "year 2000" here was just wrong. We meant "in the mid-nineties", and changed the text accordingly.

**SC36.** *Line 435: what is the availability of the observational data shown in Fig. 10 and 11?*
There is no new observational data set compiled for this paper. All observational data shown here has already been published elsewhere, with related data availability statements (e.g., the data in Fig. 10a was compiled by Waugh and Hall (2002), the data in Fig. 10b published by Engel et al. (2017), and the data in Fig. 11 by Laube et al. (2020)).

**Typos, wording, etc.:**

**L4:** *Line 4: ERA5-based results are compared to those using the preceding ERAInterim reanalysis.*
Changed to "based on the preceding..."

**L34:** *Line 34: Waugh and Hall (2002) is a review paper on this diagnostic, hence the e.g. is not necessary.*
Changed.

**L101:** *Line 101: ...a pulse period of 2 months.*
We think "frequency" is more appropriate here, as this text refers to the frequency that a pulse occurs and not to the period of one specific pulse (which would be 10 years).

**L153:** *Line 153: For instance, the forecast data starting at 6UTC and on the 5th hour forecast step.... Same for line 161.*
As the text refers to "forecast data" as a data set provided by the ECMWF at a specific time and not to the "forecast" itself, we think the formulation is more appropriate as is.

**L157:** *Line 157: Here the hourly ERA5 data is not used to drive model transport but data sub- sampled in time is used instead, hence the temperature tendencies...*
Changed accordingly.

L178: *Line 178: where ERA5 shows upward velocities whereas they are downward in ERA Interim.*
Changed as suggested.

L205: *Line 205: ...apparent in both reanalyses.*
Corrected.

L207: *Line 207: ...critically...*
Corrected.

L220: *Line 220: abbreviation NH not yet defined.*
NH is defined now in the introduction.

L231: *Line 231: clear detailed differences occur in the details of the spectrum shape*
Having "details" twice sounds somewhat complicated to us, hence we keep the formulation as is.

L246: *Line 246: However above about 500 K and in particular in the NH, the signs of the trends...*
As the trends are only opposite in the NH we think the existing formulation is more appropriate.

L286: *Line 286: ...shows increasing RCTTs, clearest with strongest positive trends in the NH.*
As the text should say that "increasing RCTTs are clearest in the NH" we think it is more appropriate as is.

L290: *Line 290: ...(compare with Fig.6).*
Changed as suggested.

L293: *Line 293: ...the variability periodicity is similar at all locations... or maybe the timing of the variability is similar at alo locations*
Here, we only want to do a qualitative comparison (sentence starts already with "From a qualitative point of view..."), and we don not want to have the formulation too specific to suggest more.

L309: *Line 309: The latitudinal distribution in ERA-Interim (Fig. 10a) agrees well with...*
Changed as suggested.

L372: *Line 372: ... this accelaration of the residual circulation...*
Changed.

---

## Author Response (AR2)

**Reply to Editor Peter Haynes**

Dear Editor Peter Haynes,

Thanks for these final comments which we think do indeed further strengthen the message of the paper. We addressed all of them in the following way:

1.) Abstract:

After discussion among the Co-authors we decided to slightly strengthen the formulation in the abstract.

"At 20km and in the NH stratosphere, ERA5 age values are at the upper margin of the uncertainty range from tracer observations, indicating a comparatively slow BDC." → "At 20km and in the NH stratosphere, ERA5 age values are around the upper margin of the uncertainty range from historical tracer observations, indicating a somewhat slow-biased BDC."

Being even stronger in this statement about a potential slow-bias of the BDC in ERA5 would be inadequate in our opinion, as the considered measurement data is to localized in time and space and associated with a too large uncertainty. For this reason, we also refrain from including a related final sentence in the abstract as proposed.

2.) Conclusions:

We adopted the text change almost as proposed, to state the slightly slow-biased BDC in ERA5 more explicitly.

"At higher stratospheric levels above the TTL and in the NH, on the other hand, ERA5 mean age is found to be at the upper margin of the observational uncertainty range." → "At higher stratospheric levels above the TTL and in the NH, on the other hand, ERA5 mean age is found to be biased somewhat high relative to observational estimates and ERA-Interim is biased slightly low, but within the uncertainly associated with the data presented. The high bias of ERA5 is, if anything, larger than the low bias of ERA-Interim."

Related to the above wording changes, we also modified the final sentence accordingly:

"However, whether the representation of stratospheric transport is indeed improved compared to ERA–Interim reanalysis is, so far, unclear." → "However, the presented comparison to observationally–based age indicates a slow-biased BDC in ERA5, although further investigations would be needed to determine the global significance of this bias."

3.) Regression method description (MC3):

We agree that adding this sentence could only enhance clarity about the method and we included it (in Sect. 4, first paragraph): "The regression did not attempt to extract any signals of interannual variability such as QBO, ENSO or volcanic influence."